# WaterSearch: A Quality-Aware Search-based Watermarking Framework for Large Language Models

## Abstract

Watermarking acts as a critical safeguard for ensuring accountability, authenticity, and trust in text generated by Large Language Models (LLMs). By embedding identifiable signals into model outputs, watermarking enables reliable attribution and enhances the security of machine-generated content. Existing approaches typically embed signals by manipulating token generation probabilities, with detection achieved by computation of corresponding statistical metrics. Despite their effectiveness, these methods inherently face a trade-off between detectability and text quality: the signal strength and randomness required for robust watermarking tend to degrade the performance of downstream tasks.

In this paper, we design a novel embedding scheme that controls seed pools to facilitate diverse parallel generation of watermarked text. Based on that scheme, we propose WaterSearch, a sentence-level, search-based watermarking framework adaptable to a wide range of existing methods. WaterSearch enhances text quality by jointly optimizing two key aspects: 1) distribution fidelity and 2) watermark signal characteristics. Furthermore, WaterSearch is complemented by a sentence-level detection method with strong attack robustness. We evaluate our method on three popular LLMs across ten diverse tasks. Extensive experiments demonstrate that our method achieves an average performance improvement of 51.01% over state-of-the-art baselines at a watermark detectability strength of 95%. In challenging scenarios such as short text generation and low-entropy output generation, our method yields performance gains of 47.78% and 36.47%, respectively. Moreover, under different attack senarios including insertion, synonym substitution and paraphrase attasks, WaterSearch maintains high detectability, further validating its robust anti-attack capabilities.

## 1 Introduction

Large language models (LLMs) demonstrate remarkable capabilities in generating high-quality content across various domains (Yang et al., 2025). However, their increasingly human-like outputs have raised growing concerns about misuse, such as fake news propagation (Vykopal et al., 2024), harmful content creation (Zugecova et al., 2024), and copyright infringement (Henderson et al., 2023). These issues highlight the urgent need for reliable detection mechanisms. As a practical solution, watermarking has emerged as a popular approach for controlling and identifying LLM-generated content. Compared with passive detection methods, watermarking offers advantages such as low computational overhead, applicability in black-box scenarios, and strong empirical detectability (Liu et al., 2024).

A common strategy for watermarking LLM-generated text is to inject statistical signals by perturbing the token-level probability distribution during generation. For example, KGW framework injects watermarks by manipulating the probability distribution of a subset of vocabulary (Kirchenbauer et al., 2023a;b; Zhao et al., 2023). Although such approaches typically preserve surface-level fluency as measured by perplexity or LLM-as-Judge metrics, empirical studies have shown that they can substantially degrade performance in downstream tasks, such as short-text (Ajith et al., 2024), reasoning (Chen et al., 2024) and low-entropy scenarios (Lee et al., 2024). Recent advancements have sought to enhance text quality by dynamically adjusting watermark strength through entropy-

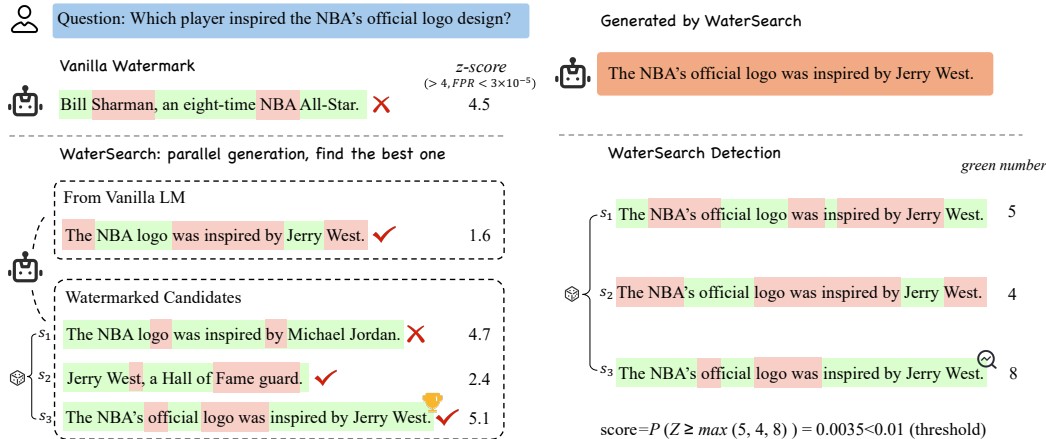

Figure 1: The mechanism and advantage of WaterSearch. Though the baseline successfully injects a watermark, it compromises text truthfulness (Top Left). In contrast, WaterSearch first choose three seeds and then generate one standard and three watermarked predictions. After that, the best response is selected by a weighted selection metric to balance generation quality and statistical detectability (Bottom Left). For detection, seeds are recover to compute the watermark probability (Right).

aware heuristics or unbiased probability expectations (Aaronson & Kirchner, 2023; Hu et al., 2023). Despite these refinements, these methods remain limited by their reliance on probability-level perturbations: any alteration of the underlying token distribution inevitably introduces stochastic deviations from the model's natural behavior. This mechanism-level constraint makes the trade-off between watermark detectability and text quality difficult to resolve.

We note that the embedding of watermarks is determined by a seed (private key). Building on this insight, we design a novel embedding scheme that dynamically manages a seed pool during sentence generation. This approach facilitates both parallel and diverse text generation. Based on this scheme, we introduce WaterSearch, a sentence-level, search-based framework that is compatible with a wide range of existing watermarking methods. **In generation**, WaterSearch formulates candidate selection after a parallel search as a multi-criteria optimization problem. It ranks sequences according to two key factors: (1) fidelity to the unwatermarked model's distribution (quality), and (2) statistical watermark strength (detectability). The generation process leverages Key-Value cache reuse to significantly reduce computational overhead. Furthermore, we theoretically establish the connection between token-level and sentence-level optimization objectives, thereby providing additional validation for the feasibility of our approach. **In detection**, WaterSearch employs a $\chi^2$ hypothesis test to achieve chunk-level verification, which is robust against token-level perturbation attacks and paraphrase attacks.

Extensive experiments demonstrate that WaterSearch consistently surpasses recent popular methods across diverse scenarios, including a 47.78% improvement in low-entropy context generations and a 36.47% improvement in short-text generation task, showcasing its remarkable versatility. Besides, our approach exhibits strong resilience against multiple watermark attack schemes like insertion, synonym substitution and paraphrase attack.

In summary, our contributions are threefold:

- We propose WaterSearch, a novel watermarking framework that optimizes both text quality and detectability of watermarked content.

- We propose a novel watermark detection method with theoretical guarantees, which could further improve the robustness of different base methods.

- Under extensive experiments, WaterSearch achieves 51.01% performance improvement on average over baselines and demonstrates substantial gains in challenging scenarios such as short-text and low-entropy generation.

## 2 RELATED WORK

### 2.1 WATERMARKING METHODS IN LLMS

Text watermarking is an active detection technique that embeds imperceptible information into generated content, enabling reliable identification (Christ et al., 2024). In the era of large language models (LLMs), watermark injection occurs during text generation by manipulating sampling distributions or output logits. Kirchenbauer et al. (2023a) first proposed the KGW framework, which partitions the vocabulary and distributions to inject watermark signals. Subsequent works have extended this framework from different perspectives. Aaronson & Kirchner (2023) analyze watermarking through the lens of probability invariance, while Lu et al. (2024) improve detectability in low-entropy scenarios, and Kuditipudi et al. (2023) aim to enhance robustness against different types of attack schemes. Apart from token-level injection, several semantic watermarking methods are designed by controlling the distribution in the semantic embedding space (Hou et al., 2024a;b). However, sampling effectiveness limits the development of these approaches (Dabiriaghdam & Wang, 2025).

### 2.2 THE DETECTABILITY-QUALITY TRADE-OFF

Existing LLM watermarking methods face a fundamental trade-off between maintaining detection reliability and preserving text quality. Perplexity evaluation in KGW (Kirchenbauer et al., 2023a) demonstrates that the method can maintain basic coherence. The LLM-as-judge framework (Singh & Zou, 2023) evaluates multiple dimensions, revealing finer-grained degradations in watermarked text. Tu et al. (2024) and Ajith et al. (2024) focus on downstream task performance, uncovering substantial degradations in watermarked text. Recent work has attempted to resolve this dilemma from different perspectives, such as expectation consistency in sampling (Hu et al., 2023) and adaptive embedding strategies (Wang et al., 2025). However, individual token selections may still introduce noticeable bias, causing factual inconsistencies and errors. Semantic-based methods either require pre-trained embedding models (Liu et al., 2023a) or partition the semantic space (Hou et al., 2024a;b), which limits their practical applicability in real-world scenarios.

### 2.3 IMPROVE GENERATED TEXT QUALITY BY SEARCH

Search-based text generation, where multiple candidates are generated and optimally selected, has been shown to achieve significant quality improvements. The inherent parallel sampling nature of LLMs has enabled numerous innovative search methods (Wang et al., 2022). In watermarking contexts, Zhang et al. (2024) employs beam search for post-hoc editing to minimize quality degradation. WaterMax (Giboulot & Furon, 2024) introduces a novel framework that selects high-information-potential chunks during generation but suffers from uneven statistical distributions. SemStamp (Hou et al., 2024a) and K-SemStamp (Hou et al., 2024b) employ rejection sampling to control the embedding distribution of each sentence in the semantic space. SimMark (Dabiriaghdam & Wang, 2025) improves sampling effectiveness but still requires an average sampling cost of 7.1 to acquire an effective watermarked sentence. Unlike these approaches, our search framework achieves a high and controllable computational cost (less than 5) with high detection confidence.

## 3 PROBLEM DEFINITION

### 3.1 TEXT GENERATION PROCESS OF LLMS

Large language models (LLMs) generate text through autoregressive sampling from a predefined vocabulary $\mathcal{V}$. Given an input prompt $\mathbf{x} = \{x_0, \ldots, x_{N-1}\}$, the model sequentially produces output tokens $\{y_0, \ldots, y_{t-1}\}$ one position at a time. To optimize this process, modern implementations leverage key-value (KV) caching, which stores intermediate attention states $\mathbf{K}_{[:N+t-1]}$ and $\mathbf{V}_{[:N+t-1]}$ for all layers, enabling efficient parallel search algorithms.

Formally, at decoding step $t$, the model computes the next-token probability distribution as:

$$\mathbb{P}(y_t \mid \mathbf{x}, \mathbf{y}_{[:t-1]}) = \text{softmax}\big(\ell^{(t)}(\mathbf{x}, \mathbf{y}_{[:t-1]}; \mathbf{K}_{[:N+t-1]}, \mathbf{V}_{[:N+t-1]})\big), \tag{1}$$

where $\ell^{(t)}(\cdot) : \mathcal{X}^N \times \mathbf{y}^{t-1} \to \mathbb{R}^{|\mathcal{V}|}$ maps the input sequence and generated tokens to logits at step $t$, and $\text{softmax}(\cdot) : \mathbb{R}^{|\mathcal{V}|} \to \Delta^{|\mathcal{V}|-1}$ projects the logits to a probability simplex over the vocabulary $\mathcal{V}$.

## 3.2 Text Watermarking of LLMs

Watermark injection aims to embed a detectable pattern into generated text by modifying the probability distribution output by LLMs. Here we formalize the mainstream KGW framework as follows (Kirchenbauer et al., 2023a).

The watermark encoding process is controlled by two key parameters, $\gamma$ and $\delta$. During each decoding step $t$, a hash function applied to previous tokens divides the vocabulary $\mathcal{V}$ into two distinct subsets: a green list $G_t$ that receives a positive bias of magnitude $\delta$, and a red list $R_t$ that remains unmodified, as formally defined in Eq. (2). Here, $\gamma$ determines the fractional size of $G_t$ relative to $\mathcal{V}$, while $\delta$ quantifies the exact logit adjustment applied to tokens in $G_t$. This design creates a fundamental trade-off where increasing $\delta$ improves watermark detectability but may adversely affect generation quality.

$$\ell_k^{(t)} = \begin{cases} \ell_k^{(t)} + \delta, & \text{if } k \in G \\ \ell_k^{(t)}, & \text{if } k \in R. \end{cases} \tag{2}$$

For detection, the process involves statistically analyzing the generated text $\mathbf{y}$ by counting green list tokens ($|\mathbf{y}|_G$) and evaluating against the null hypothesis $H_0$ that the text was produced without following the green list rule. The detection confidence is computed via the z-statistic in Eq. (3), where $T$ represents the total number of generated tokens. When the computed z-score exceeds a predetermined threshold, $H_0$ is rejected, confirming the presence of the watermark.

$$z_{\mathbf{y}} = (|\mathbf{y}|_G - \gamma T)/(\sqrt{\gamma(1-\gamma)T}). \tag{3}$$

## 3.3 Metrics of Generated Text

There are two categories of evaluation methods to measure the proximity between candidates and the optimal reference: lexical overlap and semantic similarity. For the former, ROUGE-L stands as one of the most representative algorithms. It effectively captures sentence-level structural similarity through longest common subsequence (LCS) matching, offering advantages in robustness and computational efficiency, as formulated in Eq. (4) (Lin, 2004).

$$\mathbf{q}_{\text{ROUGE-L}}(y, \tilde{y}) = \frac{(1 + \beta^2) \cdot f_{\text{LCS}}(y, \tilde{y})^2}{m \cdot f_{\text{LCS}}(y, \tilde{y}) + \beta^2 n \cdot f_{\text{LCS}}(y, \tilde{y})} \tag{4}$$

For semantic similarity measurement, a widely used approach involves using a pretrained embedding model to obtain sentence representations, followed by computing the cosine similarity between sentence pairs to reflect the semantic proximity in the latent space, as illustrated in Eq. (5) (Reimers & Gurevych, 2019).

$$\mathbf{q}_{Embedder}(y, \tilde{y}) = \frac{\mathbf{e}(y)^\top \mathbf{e}(\tilde{y})}{\|\mathbf{e}(y)\| \cdot \|\mathbf{e}(\tilde{y})\|} \tag{5}$$

## 4 Watermarked Text Selection in Parallel Generation

Under watermarking scenarios, text quality can be evaluated through dual criteria: (1) task-solving capability for downstream applications, and (2) watermark detectability.

For the first criterion, while direct evaluation on unseen downstream tasks is infeasible, comparative analysis across parallel-generated text candidates is possible (Wang et al., 2022). Since watermarked texts inherently deviate from their non-watermarked counterparts, the optimal watermarked output should minimize such deviation. Regarding detectability, as demonstrated in Eq. (3), the detection confidence monotonically increases with the count of green-list tokens, which prioritizes candidates with higher green-list token frequency. As shown in Equation (6), a straightforward quality metric design $\mathbf{q}(y, \tilde{y})$ at the *macroscopic* step is a linear combination of both coherence and detectability.

$$\mathbf{q}(y, \tilde{y}) = \alpha \mathbf{q}_{\text{sen-sim}}(y, \tilde{y}) + (1 - \alpha) \frac{|\tilde{y}|_G}{|\tilde{y}|} \tag{6}$$

Similarly, Wang et al. (2025) formulate the watermark applicability at the *microscopic* level as a multi-objective trade-off analysis function $\mathcal{J}$ at each generation step for optimal watermark strength selection,

$$
\begin{aligned}
\mathcal{J}(r) &= \mathcal{T}(r) + \omega \cdot \mathcal{W}(r) \\
&= \underbrace{\mathcal{M}(P, \hat{P})}_{\text{Text Quality}} + \omega \cdot \underbrace{\left[ (\hat{P}_G - \hat{P}_R) - (P_G - P_R) \right]}_{\text{Watermark Effectiveness}},
\end{aligned} \tag{7}
$$

where $r$ represents the watermark strength, $\mathcal{T}(r)$ measures the divergence between the watermarked $\hat{P}$ and the original $P$ probability distributions, and $\mathcal{W}(r)$ quantifies the watermark's detectability through the distributional shift.

**Theorem 1.** *(Proof in Appendix B) The microscopic objective $\max_{r} \mathcal{J}(r) = \mathcal{T}(r) + \omega \cdot \mathcal{W}(r)$ and macroscopic objective $\max_{r} \mathbb{E}[\mathbf{q}(y, \tilde{y})] = \alpha \mathbf{q}_{sen\text{-}sim}(y, \tilde{y}) + (1 - \alpha) \frac{|\tilde{y}|_G}{|\tilde{y}|}$ have the same maximizer $r^*$ when the weight is chosen as $\omega = \frac{1-\alpha}{2\alpha} \cdot \frac{1}{f'(\mathcal{T}(r))}$, where $f(\cdot)$ is a mapping that transformers a probability distribution-level similarity metric $\mathcal{T}(r)$ into a text semantic-level similarity measure.*

Theorem 1 demonstrates that macroscopic-level and microscopic-level approaches share the same optimization target, which means the sentence-level search method provides an effective approximation to token-level watermark embedding, enabling to balance quality and detectability trade-off at the sentence level. Furthermore, the selection process introduced by WaterSearch mitigates the issues of myopia and deviation in the token selection process and enables a semantic-level evaluation of text quality.

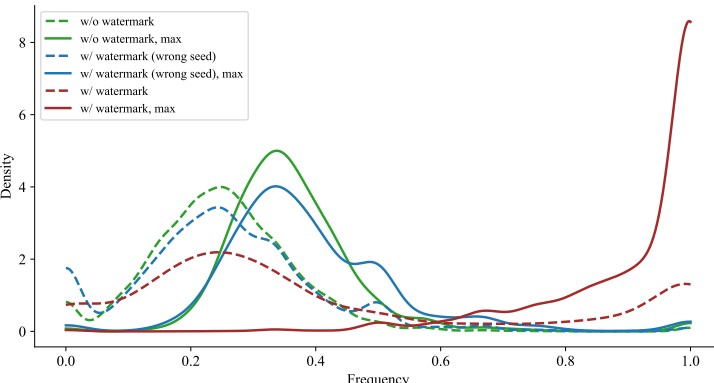

Figure 2: Frequency of token number in green list varying in different seeds under WaterSearch settings. Text without watermark shares a similar distribution with watermarked text in wrong seeds. While under correct seed shows a totally different distribution.

## 5 WATERSEARCH: A SEARCH-BASED WATERMARKING FRAMEWORK

### 5.1 GENERATION

Building upon previous discussions, we propose a search-based watermarking framework named WaterSearch. Algorithm 1 illustrates the watermark embedding process. Given an input prompt $\mathbf{c}$, WaterSearch generates an unwatermarked chunk $\mathbf{y}_i^{(0)}$ and $k-1$ watermarked chunks $[\mathbf{y}_i^{(1)}, \cdots, \mathbf{y}_i^{(k-1)}]$ in one batch. The watermark processor for each chunk can be any token-level watermarking method, with a unique hash key derived from the previous chunk. The optimal sentence $\hat{\mathbf{y}}_i$ is selected from Equation (6) to balance quality and detectability. Then the optimal chunk is concatenated to the rear of the context $\mathbf{c}$ as the input for the next generation round. The left panel of Fig. 1 demonstrates the selection process, where the best candidate is preserved and those hard to detect or inconsistent with the original text are filtered out. Seed generation process and pseudo-Code of generation could be found in Appendix A.

---

**Algorithm 1** WaterSearch Generation

---

1: **Input:** prompt $\mathbf{c} = [c_0 \cdots c_{N-1}]$, chunk size $m$, beam size $k > 1$, base logit processor $\ell$ and watermark processor $\tilde{\ell}(s_i)$ under seed $s_i$.
2: **for** $i = 0, 1, \cdots, \lceil T/m \rceil$ **do**
3:    $[s_1, \cdots, s_{k-1}]$ = RandomSeedGenerator(context, k-1)    $\triangleright$ Generate k-1 seeds by context information.
4:    Initialize watermark logitprocessor list by seeds:
   $\ell_{WS} = [\ell, \tilde{\ell}(s_1), \cdots, \tilde{\ell}(s_{k-1})]$      $\triangleright$ Build logit processors by Eq. 2 with different seeds.
5:    $[\mathbf{y}_i^{(0)}, \mathbf{y}_i^{(1)}, \cdots, \mathbf{y}_i^{(k-1)}]$=ParallelGeneration($\mathbf{c}, \ell_{WS}$)    $\triangleright$ Parallel generation with the same prefix $\mathbf{c}$.
6:    Choose response $\hat{\mathbf{y}}_i$ by Eq. (6)
7:    $\mathbf{c} = [\mathbf{c}||\hat{\mathbf{y}}_i]$        $\triangleright$ Reuse KV-cache.
8: **end for**
9: **Output:** watermarked text $[\hat{\mathbf{y}}_1 \cdots, \hat{\mathbf{y}}_{\lceil T/m \rceil}]$

---

**Algorithm 2** WaterSearch Detection

---

1: **Input:** prompt $\mathbf{c} = [c_0 \cdots c_{N-1}]$, chunk size $m$, beam size $k$, response $\mathbf{y} = [y_0, \cdots, y_{T-1}]$.
2: PvalueList = []
3: split $\mathbf{y}$ by chunk size $m$ to $[\mathbf{y}_1 \cdots, \mathbf{y}_{\lceil T/m \rceil}]$.
4: **for** $\mathbf{y}_i$ in $\mathbf{y}$ **do**
5:    seeds=RandomSeedGenerator(context, k-1)      $\triangleright$ Recover the seeds from generation.
6:    Pvalue=ChunkSignificantTest($\mathbf{y}_i$,seeds)    $\triangleright$ Calculate p-value of each chunk by Eq. 9 and Eq. 10.
7:    PvalueList.append(Pvalue)
8: **end for**
9: DocPvalue=DocSignificantTest(PvalueList)    $\triangleright$ Calcuate the overall confidence by Eq. 11.
10: **Output:** DocPvalue

---

## 5.2 DETECTION

Given a text generated by WaterSearch, it is hard to identify the exact seed that matches the watermark processor of the selected chunk. However, a huge deviation of statistical significance is observed between the correct seed and incorrect seeds for the texts. As shown in Fig. 2, if a wrong key is applied to the text (blue dashed line), no statistical significance is observed which ends up like normal text (green dashed line). To design a detection algorithm, we first design a hypothesis test as follows:

$H_0$: *The text sequence is generated with no knowledge of parallel watermarking selection.*

$H_1$: *The text sequence is generated by WaterSearch.*

Under $H_0$ settings, denote $z_i(s)$ as the green token number $|\mathbf{y}_i|_{G(s)}$ of the $i$-th chunk $\mathbf{y}_i$ under a seed $s$, then $z(s)$ follows a binomial distribution $z(s) \sim \mathcal{B}(|\mathbf{y}_i|, \gamma)$. Then, we define maximum of $z(s)$ under $[s_1, \cdots, s_m]$:

$$Z_i = \max\left(z_i(s_1), \cdots, z_i(s_m)\right), \; z_i(s_j) \sim B(|\mathbf{y}|, \gamma). \tag{8}$$

$Z_i$ follows the maximum of multiple independent binomial distributions, thus the CDF of $Z$ is derived as:

$$F(Z_i) = P(Z_i \le z) = \left[\sum_{j=0}^{z} \binom{n}{j} p^j (1-p)^{n-j}\right]^m. \tag{9}$$

The corresponding p-value is expressed by Eq. (10), where $z_{\text{obs}}$ denotes the observed maximum value in the sample. Apart from KGW framework, the detection process could hold as well if the chunk-level p-value in Eq. (10) can be calculated or expressed approximately. The statistical significance between $H_0$ and $H_1$ is shown between brown solid line and dashed line of Fig. 2.

$$p_i = P(Z_i \ge z_{\text{obs}}) = 1 - F(z_{\text{obs}} - 1) \tag{10}$$

$$F_{\chi^2} = -2 \sum_{i=1}^{\lceil T/m \rceil} \ln(p_i) \sim \chi^2\left(2\lceil T/m \rceil\right) \tag{11}$$

After that, we acquire a set of p-values $\left[p_1, \cdots, p_{\lceil T/m \rceil}\right]$ from $\lceil T/m \rceil$ chunks. Then, Fisher's combined probability test for aggregating a set of p-values is introduced. Fisher's test comprehensively considers the watermark likelihood across different segments and provides reliable judgments. Even if individual tests show weak effects, the combined result may still achieve significance when multiple tests consistently show significant results. The overall confidence to reject $H_0$ can be computed in Eq. (11). Pseudo-Code of detection could be found in Appendix A.

# 6 EXPERIMENTS

## 6.1 EXPERIMENTAL SETUP

We choose WaterBench (Tu et al., 2024) as our evaluation benchmark. WaterBench is a multi-task benchmark spanning nine typical NLP tasks with varying input / output lengths. Moreover, it requires a consistent watermarking strength for a fair comparison. We also add the RepoBench-P (Liu et al., 2023b) dataset to supplement the dataset for code scenarios. Following these settings, we evaluate on three representative LLMs: Qwen-2.5-7B-Instruct (Yang et al., 2025), Llama-2-7B-chat (Touvron et al., 2023) and InternLM-7B-chat (InternLM Team, 2023). Four representative watermarks are included: KGW-Hard (Kirchenbauer et al., 2023a), KGW-Soft (Kirchenbauer et al., 2023a), GPT Watermark (Zhao et al., 2023) and V2 Watermark (Kirchenbauer et al., 2023b). During the evaluation, watermark strength (True Positive Rate, TPR) is set to 95% for pairwise comparison on different downstream tasks performance.

For WaterSearch, though semantic-based methods (Eq. 5) may yield a better similarity metric, we employ ROUGE-L for its low computation cost (Appendix D.3 compares the 2 types of metrics). For hyper-parameters, we set the number of beams $m$ to 5, consisting of 1 standard text without watermark and 4 watermarked texts with distinct seeds. For detection, we set confidence threshold for the p-value to 0.01. Details of parameter selection are discussed in Appendix D.1.

## 6.2 MAIN RESULTS

| | Method | Short In/Short Out | | Short In/Long Out | | Long In/Short Out | | | Long In/Long Out | | Open-Ended | Overall | | |
|---|---|---|---|---|---|---|---|---|---|---|---|---|---|---|
| | | KoLA | Copen | ELI5 | FiQA | HotpotQA | LCC | RepoBench-P | Multinews | QMsum | AlpacaFarm | TP | TN | GM |
| Qwen-2.5-7B-Instruct | Hard Watermark | 6.2 | 4.5 | 20.0 | 17.3 | 13.6 | 20.3 | 15.4 | 17.2 | 14.2 | 13.3 | 95.6 | 99.5 | 14.2 |
| | + WaterSearch | 13.3 | 37.8 | 22.6 | 18.7 | 29.3 | 22.6 | 25.1 | 19.7 | 18.0 | 27.7 | 96.4 | 99.7 | **23.5** |
| | Soft Watermark | 3.1 | 41.0 | 16.5 | 14.1 | 36.1 | 15.2 | 17.4 | 12.2 | 14.4 | 6.4 | 94.8 | 99.7 | 17.6 |
| | + WaterSearch | 9.9 | 41.0 | 19.1 | 15.0 | 45.9 | 20.2 | 23.6 | 13.5 | 15.8 | 11.1 | 96.4 | 99.8 | **21.5** |
| | GPT Watermark | 4.6 | 42.0 | 5.6 | 5.5 | 35.3 | 5.0 | 8.4 | 3.6 | 11.7 | 4.4 | 95.3 | 99.7 | 12.6 |
| | + WaterSearch | 7.9 | 40.8 | 14.9 | 13.0 | 40.0 | 17.8 | 16.8 | 10.8 | 15.2 | 7.1 | 97.3 | 99.1 | **18.4** |
| | V2 Watermark | 5.1 | 23.3 | 19.3 | 15.5 | 16.6 | 12.5 | 13.5 | 13.5 | 13.7 | 7.1 | 95.5 | 99.8 | 14.0 |
| | + WaterSearch | 5.4 | 38.0 | 21.7 | 17.7 | 32.0 | 22.0 | 22.2 | 17.2 | 16.3 | 20.6 | 98.9 | 99.7 | **21.3** |
| Llama2-7B-Chat | Hard Watermark | 1.1 | 8.9 | 10.5 | 13.6 | 4.9 | 27.8 | 25.9 | 11.1 | 12.2 | 1.1 | 95.6 | 99.5 | 11.7 |
| | + WaterSearch | 2.4 | 28.0 | 13.7 | 16.4 | 8.6 | 29.0 | 27.2 | 15.0 | 14.0 | 4.5 | 98.7 | 99.5 | **15.9** |
| | Soft Watermark | 1.7 | 13.8 | 8.1 | 11.8 | 14.4 | 25.3 | 22.2 | 9.3 | 11.0 | 0.6 | 95.3 | 99.5 | 11.8 |
| | + WaterSearch | 2.4 | 27.2 | 12.4 | 16.0 | 16.3 | 27.6 | 27.1 | 13.6 | 13.5 | 4.5 | 97.6 | 99.6 | **16.1** |
| | GPT Watermark | 1.8 | 25.3 | 4.5 | 5.9 | 12.5 | 17.0 | 12.9 | 4.8 | 9.6 | 0.2 | 97.0 | 96.9 | 9.5 |
| | + WaterSearch | 1.6 | 31.0 | 11.1 | 13.5 | 17.4 | 21.2 | 21.9 | 11.0 | 13.2 | 4.5 | 96.5 | 99.0 | **14.6** |
| | V2 Watermark | 1.1 | 21.3 | 13.2 | 13.5 | 7.4 | 20.4 | 23.3 | 11.7 | 11.5 | 0.9 | 94.5 | 99.9 | 12.4 |
| | + WaterSearch | 2.3 | 33.6 | 14.6 | 16.8 | 15.2 | 27.8 | 27.8 | 16.2 | 13.9 | 5 | 99.4 | 99.8 | **17.3** |
| InternLM-7B-Chat | Hard Watermark | 2.8 | 0.8 | 10.7 | 8.4 | 3.2 | 20.1 | 15.8 | 5.3 | 7.4 | 0.8 | 93.3 | 99.7 | 7.5 |
| | + WaterSearch | 2.6 | 6.1 | 13.8 | 12.9 | 3.5 | 20.9 | 20.7 | 8.4 | 10.6 | 2.0 | 99.8 | 99.2 | **10.2** |
| | Soft Watermark | 2.4 | 10.1 | 9.1 | 6.1 | 2.5 | 18.6 | 17.7 | 4.0 | 5.3 | 0.3 | 94.0 | 99.6 | 7.6 |
| | + WaterSearch | 5.0 | 24.6 | 16.1 | 15.5 | 15.4 | 15.3 | 15.8 | 12.6 | 11.5 | 5.3 | 97.1 | 99.2 | **13.7** |
| | GPT Watermark | 1.9 | 4.5 | 8.5 | 7.1 | 2.4 | 20.5 | 19.4 | 4.2 | 6.2 | 0.5 | 95.6 | 99.8 | 7.5 |
| | + WaterSearch | 4.2 | 11.0 | 15.6 | 15.2 | 15.6 | 25.9 | 17.1 | 12.4 | 12.4 | 5.3 | 98.4 | 98.7 | **13.5** |
| | V2 Watermark | 1.3 | 20.6 | 9.0 | 6.3 | 3.4 | 28.1 | 27.3 | 5.3 | 5.6 | 0.5 | 94.9 | 99.8 | 10.7 |
| | + WaterSearch | 4.7 | 33.5 | 15.3 | 15.4 | 16.7 | 28.8 | 33.3 | 12.5 | 11.5 | 4.5 | 95.3 | 99.6 | **17.6** |

Table 1: Performance comparison of WaterSearch and counterpart base watermarking methods across 10 benchmark datasets. The final "Overall" column summarizes both generation quality (GM) and detection accuracy metrics: True Positive rate (TP, correctly judge the text has a watermark) and True Negative rate (TN, correctly judge the text does not have a watermark).

We evaluate WaterSearch against baseline watermarking methods across ten tasks. Table 1 reports the overall performance relative to their corresponding base methods. Our approach achieves an average improvement of 51.01% in downstream task performance. Notably, under InternLM-7B-Chat with the GPT Watermark, WaterSearch delivers an 80% performance gain, clearly demonstrating its effectiveness. Even the minimum improvement reaches 22.16%. These consistent gains across

Table 2: Detail results in short text and low entropy generation scenarios.

| | Short Text Generation | | | | | | | | | Low Entropy Generation | | | | | |
| | KoLA | | | Copen | | | HotpotQA | | | LCC | | | RepoBench-P | | |
| | TP | TN | GM | TP | TN | GM | TP | TN | GM | TP | TN | GM | TP | TN | GM |
|---|---|---|---|---|---|---|---|---|---|---|---|---|---|---|---|
| Hard Watermark | 100.0 | 100.0 | 1.1 | 79.0 | 100.0 | 8.9 | 72.0 | 100.0 | 4.9 | 93.0 | 100.0 | 27.8 | 97.5 | 100.0 | 25.9 |
| WaterSearch ($\alpha$=0.75) | 100.0 | 98.0 | **2.4** | 96.4 | 100.0 | **28.0** | 95.5 | 100.0 | **8.6** | 95.0 | 99.4 | **29.0** | 92.0 | 98.0 | **27.2** |

different models and dataset configurations underscore the broad applicability and robustness of our method. Considering that current mainstream watermarking techniques often underperform on short texts and in low-entropy scenarios, we provide a detailed analysis of these two challenging settings.

**Short Text Generation**  Due to the limited number of tokens in short-text generation, conventional methods often struggle to accumulate sufficient confidence for accurately detecting watermarks. Take hard watermark as an example, the method is shown in Table 2. It's worth notice that WaterSearch improve the detection success rate in Copen and HotpotQA with 22.03% and 32.64%. This could attribute to search for sentence in high green-token densities, thereby facilitating the accumulation of watermark signals.

**Low-Entropy Text Watermarking**  Low-entropy text generation presents a significant challenge. Code completion is a typical low-entropy scenario due to the inherent structural and syntactic constraints of programming languages. Tokens in code generation are highly predictable; thus, applying strong watermarks may disrupt syntax, whereas weak watermarks may not sufficiently perturb the sampling distribution. As shown in Table 2, our method maintains high text quality in low-entropy scenarios. As indicated in Eq. 6, WaterSearch mitigates quality degradation by preserving high similarity to the original, non-watermarked text. At the same time, it enhances detectability by selecting text segments exhibiting strong watermark signals.

## 6.3 ATTACKING ROBUSTNESS

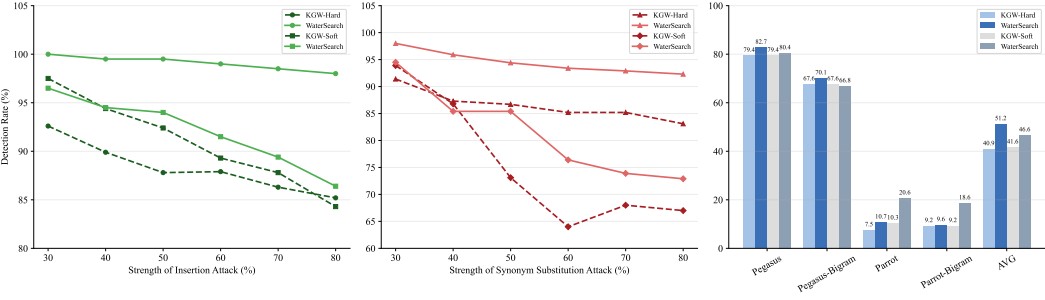

Figure 3: Detection success rate under different types of watermark attacks: Insertion attack (left), Synonym substitution attack (middle) and Paraphrase attack (right). WaterSearch exhibits robustness to its counterpart among all the attacks.

Resistance to attacks is a key metric for evaluating the practicality of a watermarking method in complex environments. In this section, we evaluate our method against two types of watermark attacks: token-level perturbation and paraphrase attack. We first generate natural language passages from the RealNewsLike subset of the C4 dataset (Raffel et al., 2020) and apply different attacks following previous work (Pan et al., 2024; Dabiriaghdam & Wang, 2025). We choose two classic token-level perturbation methods, insertion attack and synonym substitution attack, to test our method. The left and middle panels of Fig. 3 showcase the drop in detection success rate as attack strength increases from 30% to 80%. Compared to different baseline methods, WaterSearch exhibits stronger resilience against both attacks.

We further evaluate WaterSearch on paraphrase attack, which is a challenging scenario that preserves the meaning of documents but significantly alters the original text by paraphrasing sentences. Here we report the experimental results for two classic paraphrasers, Pegasus (Zhang et al., 2020) and Parrot (Damodaran, 2021), as well as the corresponding bigram paraphrase attack introduced by

Hao et al. (2025). As shown in Fig. 3 (right), our method outperforms its counterparts significantly, especially on the Parrot and Parrot-Bigram paraphrasers, achieving improvements of 10.3 and 9.4 percentage points over KGW-Soft.

The detection robustness likely stems from the rigorous assessment in $\chi^2$ detection: even for a document under strong attacks, each sentence can still retain relatively sufficient watermark signals to enable hypothesis testing. Please find full results and further discussions in Appendix E.

## 6.4 SCALING EFFECT OF PARALLEL SEARCH

This section discusses the scaling effect of parallel search, specifically focusing on the performance dynamics as the search beam width $K$ increases. We conduct experiments under two settings, lexical versus semantic similarity, and report results for text quality (GM) and detection precision (TP). Fig. 5 demonstrates the scaling effect of both GM and TP with the increase of $K$. Notably, our method outperforms the baseline ($GM = 11.8$) even when $K = 2$, and as $K$ increases, the detection accuracy gradually exceeds that of the baseline ($TP = 95.3$). Detailed experimental results are provided in Table 11 and Table 12 in Appendix F.1.

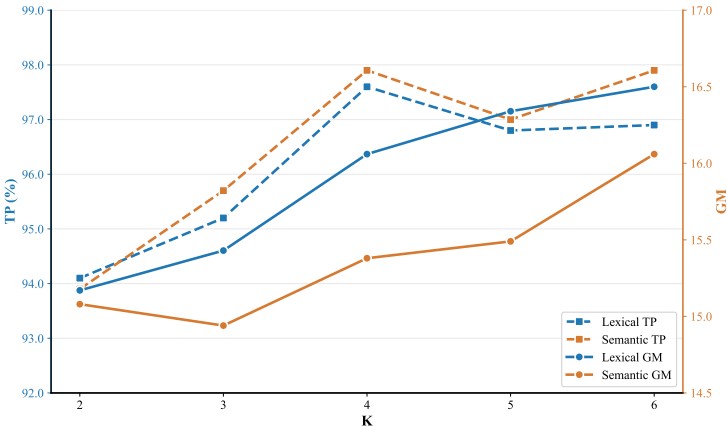

Figure 4: Scaling effect of parallel number $K$ of WaterSearch in Generation and Detection.

## 6.5 COMPUTATION COST

This section discuss the throughput and memory cost of WaterSearch. Intuitively, parallel search in $K$ beams will requires $K$ times of memory usage and deteriate throughput to $1/K$. WaterSearch's computational overhead is optimized in two aspects: (i) efficient parallel inference supported by the LLM architecture and (ii) KV-cache reuse of previous chunks for subsequent generation. This enables WaterSearch to maintain competitive throughput while achieving improved generation quality.

Table 3: Space complexity comparison.

| Inference Method | Space Complexity (Peak KV Cache Memory) |
|---|---|
| Sequential | $\mathcal{O}(L + T)$ |
| Parallel | $\mathcal{O}(k(L + T))$ |
| WaterSearch | $\mathcal{O}(k(L + m))$ |

Table 3 compares the peak KV cache memory consumption of three generation paradigms. Here input and output length are denoted as $L, T$, $k$ refers parallel generation numbers, and $m$ as chunk size, which normally much smaller than output length ($m \ll T$). It can be observed that the memory footprint of parallel generation scales linearly with $k$, leading to substantial memory consumption. In contrast, our method generates small text chunks in parallel at each step and leverages cache reuse, which is particularly advantageous for long-sequence generation.

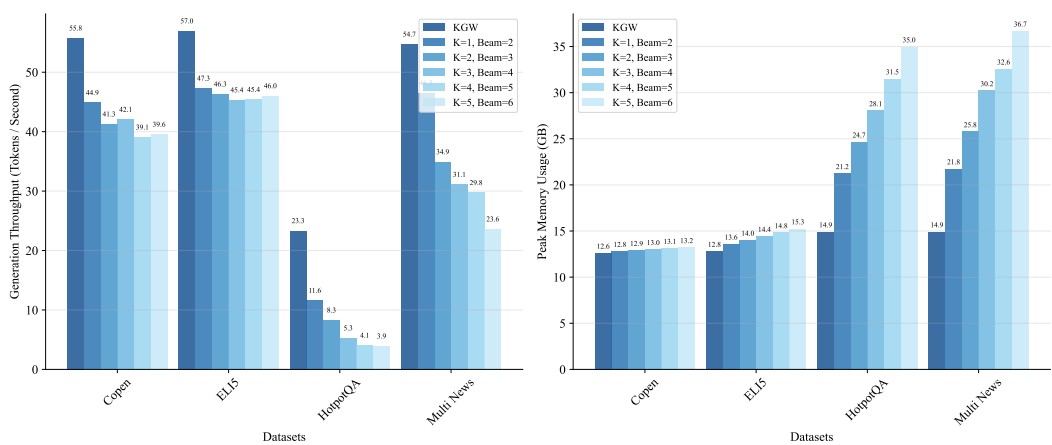

Figure 5: Throughput and Peak Memory Usage in WaterSearch

To assess the practical computational overhead, we report throughput and peak memory consumption across four datasets: Copen (short input / short output), ELI5 (short input / long output), HotpotQA (long input / short output), and MultiNews (long input / long output). All experiments are conducted on a single NVIDIA A800 GPU using HuggingFace Transformers (Wolf et al., 2020) and FlashAttention2 (Dao et al., 2022). As illustrated in Fig. 5, WaterSearch introduces additional cost, which is expected given its design. However, on short-input tasks such as Copen and ELI5, the impact on throughput is minimal and the increase in peak memory is modest. For long-input tasks, including HotpotQA and MultiNews, the overhead becomes more apparent; nonetheless, the growth remains sublinear, demonstrating that WaterSearch scales efficiently in practice. Importantly, the method achieves substantial quality gains even at small parallel degrees (e.g., $k = 2$, see Section 6.4), providing flexible hyperparameters that can be tuned to accommodate different computational budgets.

## 7 CONCLUSION

In this work, we present WaterSearch, a novel chunk-level watermarking framework for LLM-generated text. The proposed method adaptively selects text fragments to enable high quality and strong detectability of watermarked text, while remaining compatible with a wide range of existing watermarking techniques. Building upon this architecture, we develop a detection method with statistical assurance. Comprehensive experimental results demonstrate the superior performance of our approach across downstream tasks—including challenging scenarios such as low-entropy generation, short-text output, and watermark attacks—validating its effectiveness and robustness.

## ETHICS STATEMENT

This work adheres to the ICLR Code of Ethics, ensuring ethical compliance throughout all stages of the research. We conducted all experiments using open-source datasets and did not involve any personally identifiable information or sensitive content. The primary goal of our research is to alleviate the misuse of AI models—a pursuit with significant potential benefits for scientific research and engineering.

## REPRODUCIBILITY STATEMENT

To support reproducibility, we provide detailed descriptions of our metrics and framework in Section 5 and Appendix A. Additionally, we include comprehensive implementation details and hyperparameter selection guidelines in Section 6.1 and Appendix D.1. We will release our data and code following the anonymous review process. Our implementation is built upon the WaterBench codebase (Tu et al., 2024). For adversarial robustness evaluation, we adopt the token-level attack

implementation from MarkLLM (Pan et al., 2024) and the paraphrase attack codebase from Sim-Mark (Dabiriaghdam & Wang, 2025).

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

## APPENDIX

# A SEED GENERATION PROCESS AND PSEUDO-CODE OF WATERSEARCH

Seed generation process includes 2 parts, seed pool generation and token-level seed generation. The former process can be formalized as

$$Seed\_Pool = Hash([s_1, s_2, \cdots], key) = shuffle([s_1, s_2, \cdots], key), \tag{12}$$

where $[s_1, s_2, \cdots]$ is a large group of seeds, and the hash function could be a shuffle function condition of key in practice.

The token-level seed generation in the KGW framework can be expressed as:

$$Seed = Hash([x^{-h}, \cdots, x^{-2}, x^{-1}]) = (x^{-h} \cdot x^{-2} \cdot x^{-1}) \, mod \, M, \tag{13}$$

where the right side of the equation is employed in practice (Kirchenbauer et al., 2023a; Fernandez et al., 2023), and $M$ is a large prime number. **It is worth notice that Seed Pool and Seed generation process are parallel.**

Based on Eq. 12 and Eq. 13, we demonstrate the generation and detection of WaterSearch in Algorithm 3 and Algorithm 4 for better understanding.

---

**Algorithm 3** Pseudo-Code in WaterSearch Generation

---

1: random.seed(41)  ▷ pre-defined secret key
2: $seed\_pool \leftarrow [1, \ldots, 5000]$  ▷ pre-defined seed pool
3: **for** $t = 1$ to $L/m$ **do**
4:  ▷ iterate over each chunk
5:  random.shuffle($seed\_pool$)  ▷ build seed pool by Eq. 12
6:  $cur\_seeds \leftarrow seed\_pool[1:K]$
7:  $no\_wm\_chunk \leftarrow model.generate(input\_text\_ids)$  ▷ generate a no watermarked chunk for reference
8:  $wm\_chunks \leftarrow []$
9:  **for** $i = 1$ to $K$ **do**
10:   ▷ iterate over each beam (parallel in practice)
11:    $token\_seed \leftarrow hash\_func(input\_text\_ids[-h:], cur\_seeds[i])$ ▷ generate chunk seed by Eq. 13
12:    $wm\_chunk \leftarrow model.generate(input\_text\_ids, wm\_logit\_processor, token\_seed)$
13:    append $wm\_chunk$ to $wm\_chunks$
14:  **end for**
15:  $idx \leftarrow calculate\_best\_wm\_chunk(no\_wm\_chunk, wm\_chunks)$  ▷ choose the best candidate by Eq. 6
16:  $input\_text\_ids \leftarrow concat(input\_text\_ids, wm\_chunks[idx])$  ▷ concat previous and current text and key-value cache for next generation
17: **end for**
18: **Output:** tokenizer.decode($input\_text\_ids$)

---

**Algorithm 4** Pseudo-Code in WaterSearch

---

1: random.seed(41)  ▷ config secret key in generation
2: $seed\_pool \leftarrow [1, \ldots, 5000]$  ▷ config seed pool in generation
3: $chunk\_wm\_scores \leftarrow []$
4: **for** $i = 1$ to $L/m$ **do**
5:  ▷ iterate over each chunk
6:  random.shuffle($seed\_pool$)  ▷ rebuild seed pool by Eq. 12
7:  $cur\_seeds \leftarrow seed\_pool[1:K]$  ▷ recover seeds in generation
8:  $chunk\_text \leftarrow output\_text[(i-1)m : im]$
9:  $chunk\_wm\_score \leftarrow chunk\_detect(chunk\_text, cur\_seeds)$  ▷ chunk-level detection in Eq. 10
10:  append $chunk\_wm\_score$ to $chunk\_wm\_scores$
11: **end for**
12: $p\_value \leftarrow document\_detect(chunk\_wm\_scores)$  ▷ document-level detection via $\chi^2$ test in Eq. 11
13: **Output:** $p\_value$

---

## B  THEORETICAL DISCUSSIONS

### B.1  PROOF OF THEOREM 1

**Theorem 2.** The microscopic objective $\max_r \mathcal{J}(r) = \mathcal{T}(r) + \omega \cdot \mathcal{W}(r)$ and macroscopic objective $\max_r \mathbb{E}[\mathbf{q}(y, \tilde{y})] = \alpha \mathbf{q}_{\text{sen-sim}}(y, \tilde{y}) + (1 - \alpha)\frac{|\tilde{y}|_G}{|\tilde{y}|}$ have the same maximizer $r^*$ when the weight is chosen as $\omega = \frac{1-\alpha}{2\alpha} \cdot \frac{1}{f'(\mathcal{T}(r))}$, where $f(\cdot)$ is a mapping that transformers a probability distribution-level similarity metric $\mathcal{T}(r)$ into a text semantic-level similarity measure.

**Preliminary.** From *microscopic* settings, vocabulary can be split into a green list $\mathcal{V}_G$ and a red list $\mathcal{V}_R$. Let $P_G = \sum_{j \in G} p_j$ represent the sum of probabilities of green tokens, and the total increase of $P_G$ as $r \cdot (1 - P_G)$, where $r$ is used to represent watermark strength and $r \in (0, 1)$. Then, we have the watermarked sampling distribution $\hat{P} = \{\hat{p}_i\}^{|\mathcal{V}|}$:

$$\hat{p}_i = \begin{cases} \frac{p_i e^\delta}{\sum_{j \in G} p_j e^\delta + \sum_{j \in R} p_j}, & \mathcal{V}_i \in \mathcal{V}_G, \\ \frac{p_i}{\sum_{j \in G} p_j e^\delta + \sum_{j \in R} p_j}, & \mathcal{V}_i \in \mathcal{V}_R. \end{cases} \tag{14}$$

Watermark effectiveness of the watermark can be quantified by the difference between the adjusted probabilities of tokens in the green list and those in the red list and given by:

$$\begin{aligned} \mathcal{W}(r) &= (\hat{P}_G - \hat{P}_R) - (P_G - P_R) \\ &= 2r(1 - P_G). \end{aligned} \tag{15}$$

Then, a multi-objective trade-off analysis function $\mathcal{J}$ as a weighted sum of text quality and watermark effectiveness:

$$\begin{aligned} \mathcal{J}(r) &= \mathcal{T}(r) + \omega \cdot \mathcal{W}(r) \\ &= \mathcal{T}(r) + \omega \cdot 2r(1 - P_G). \end{aligned} \tag{16}$$

From *macroscopic* settings, we make several assumptions and declarations for proving convenience:

1. Token Independence: The watermarked text $\tilde{y} = (t_1, \cdots, t_n)$ is generated by sampling each token independently from the perturbed distribution $\hat{P}$.

2. Green List Proportion: The macroscopic detectability term $\frac{|\tilde{y}_G|}{|\tilde{y}|}$ is the empirical fraction of green-list tokens, whose expectation is $\hat{P}_G$.

3. Semantic Similarity Proxy: The expected macroscopic semantic similarity $\mathbb{E}[\mathbf{q}_{sen-sim}(y, \tilde{y})]$ is a differentiable, strictly increasing function $f$ of $\mathcal{T}(r)$.

**Proof.** By assumption 2 and 3, we have,

$$\mathbb{E}\left[\frac{|\tilde{y}_G|}{|\tilde{y}|}\right] = \hat{P}_G, \tag{17}$$

and

$$\mathbb{E}[\mathbf{q}_{sen-sim}] = f(\mathcal{T}(r)). \tag{18}$$

To find the optimal $r^*$, do partial derivative to macroscopic objective, we have

$$\frac{d}{dr}\mathbb{E}[\mathbf{q}] = \alpha f'(\mathcal{T}(r))\mathcal{T}'(r) + (1 - \alpha)\frac{d\hat{P}_G}{dr} = 0. \tag{19}$$

Do derivative to microscopic objective, we have

$$\mathcal{J}'(r) = \mathcal{T}'(r) + \omega \mathcal{W}'(r) = 0 \tag{20}$$

By Eq. 20,

$$\omega = -\frac{\mathcal{T}'(r^*)}{\mathcal{W}'(r^*)}. \tag{21}$$

By Eq. 19,

$$\mathcal{T}'(r^*) = -\frac{1-\alpha}{\alpha f'(\mathcal{T}(\nabla^*))} \frac{d\hat{P}_G}{dr} \tag{22}$$

Replace Eq. 21 with Eq. 22,

$$\omega = \frac{1-\alpha}{\alpha f'(\mathcal{T}(\nabla^*))} \cdot \frac{d\hat{P}_G/dr}{\mathcal{W}'(r^*)} = \frac{1-\alpha}{\alpha f'(\mathcal{T}(\nabla^*))} \cdot \frac{\partial \hat{P}_G}{\partial \mathcal{W}(r^*)} \tag{23}$$

Since $\hat{P}_G = \sum_{i \in V_G} \hat{p}_i = \sum_{i \in V_G} p_i \left( 1 + \frac{r(1-P_G)}{P_G} \right) = P_G + r(1 - P_G)$ and $\mathcal{W}(r) = 2r(1 - P_G)$, then

$$\frac{d\hat{P}_G}{dr} = 1 - P_G, \quad \frac{d\mathcal{W}}{dr} = 2(1 - P_G). \tag{24}$$

Then,

$$\omega = \frac{1-\alpha}{\alpha f'(\mathcal{T}(\nabla^*))} \cdot \frac{\partial \hat{P}_G}{\partial \mathcal{W}(r^*)} = \frac{1-\alpha}{2\alpha} \cdot \frac{1}{f'(\mathcal{T}(\nabla^*))} \tag{25}$$

If $f'(\mathcal{T}(r))$ is constant near $r^*$, then $\omega$ is irrelevant to $r^*$.

When $f(x) = x$, then $f' = 1$, $\omega = \frac{1-\alpha}{2\alpha}$. This completes the proof.

## C DATASET DETAILS

The information of all the datasets is shown in Table 4, which categorizes based on the length of input or output. Apart from WaterBench (Tu et al., 2024), we additionally introduce RepoBench-P (Liu et al., 2023b) to complete evaluations in low-entropy scenarios.

Table 4: Details of all the experiment datasets.

| Category | Dataset | Task | Metric | Avg.Length |
|---|---|---|---|---|
| *Short In/Short Out* | KoLA (Yu et al., 2024) | Entity Probing | F1 | 11.9/5.9 |
| | Copen (Peng et al., 2022) | Concept Probing | F1 | 84.1/3.9 |
| *Short In/Long Out* | ELI5 (Fan et al., 2019) | Long-form QA | ROUGE-L | 49.4/277.2 |
| | FiQA (Maia et al., 2018) | Finance QA | ROUGE-L | 17.9/302.2 |
| *Long In/Short Out* | HotpotQA (Yang et al., 2018) | Multi-Doc QA | F1 | 15266.0/5.6 |
| | LCC (Chen et al., 2021) | Code Completion | Edit Distance | 4183.8/16.8 |
| | RepoBench-P (Liu et al., 2023c) | Code Completion | Edit Distance | 14696.8/18.9 |
| *Long In/Long Out* | Multinews (Fabbri et al., 2019) | Multi-Doc Summary | ROUGE-L | 3114.4/329.7 |
| | QMsum (Zhong et al., 2021) | Query-Based Summary | ROUGE-L | 15923.1/87.9 |
| *Open-Ended Generation* | AlpacaFarm (Dubois et al., 2024) | Instruction Following | GPT-4 Judge | 41.0/86.2 |
| | C4(Raffel et al., 2020) | General Writing | Perplexity | 35.4/233.2 |

# D    MORE EXPERIMENTAL RESULTS

## D.1    DETAILED EXPERIMENTAL SETUPS

For the of baseline method in Table 1, we report the result in (Tu et al., 2024). Considering the timeliness of the models, we have added Qwen-2.5-Instruct as a supplement. Aligned with Water-Bench settings, the $\gamma$, $\delta$ hyper-paramters are acquired by grid search to reach 0.95 True Positive Rate. We first initialize the hyper-parameters for the Qwen-2.5 model experiments using those from the Llama-2 model experiments. As an increase in $\gamma$ leads to weaker watermark strength and an increase in $\delta$ results in stronger watermarking, we leveraged this relationship during grid search to find the hyper-parameter setting that minimizes the smallest deviation from the target TPR. The full settings are shown in 5.

| Method / Params | Llama2-7B-Chat | InternLM-7B-Chat | Qwen2.5-7B-Instruct |
|---|---|---|---|
| Hard Watermark ($\gamma$) | 0.25 | 0.15 | 0.35 |
| Soft Watermark ($\gamma/\delta$) | 0.1/10 | 0.1/10 | 0.1/12 |
| GPT Watermark ($\gamma/\delta$) | 0.1/10 | 0.25/15 | 0.1/12 |
| V2 Watermark ($\gamma/\delta$) | 0.25/15 | 0.1/10 | 0.25/15 |

Table 5: Hyper-parameters settings of baseline methods

As for WaterSearch method, since the watermarking strength is stronger than KGW-based methods, we fixed $\gamma$ the same as counterpart, and decrease $\delta$ to reach the overall 95% TPR. After that, we turn $\alpha$ for a better performance, which is further discussed in Appendix F.3.

## D.2    EXPERIMENTAL RESULTS ON MORE MODELS

We evaluate WaterSearch on more models, including state-of-the-art models Qwen3-14B and Qwen3-32B (Yang et al., 2025). Here we set $\gamma = 0.1, \delta = 12.0$ in KGW-Soft and $\gamma = 0.05, \delta = 12.0$ in WaterSearch to assure the watermarking strength around 95%. The results are shown in Table 6 and Table 7, which demonstrate the effectiveness across diverse models. During experiment, we notice the strong LLMs may possess a sharp logit distribution, which worths for further discovering. Limited by the maximum memory of single graphics card, we only report the result of Qwen3-14B in 4K and Qwen3-32B in 3K context length. We will support multi-device implementation to faciliate LLMs in larger scale.

Table 6: Comparison of WaterSearch and KGW-soft on Qwen-3-14B in 4K context length.

| Method | Short In/Short Out | | Short In/Long Out | | Long In/Short Out | | | Long In/Long Out | | Open-Ended | Overall | |
|---|---|---|---|---|---|---|---|---|---|---|---|---|
| | KoLA | Copen | ELI5 | FiQA | HotpotQA | LCC | RepoBench-P | Multinews | QMsum | AlpacaFarm | TP | GM |
| KGW-Soft | 12.6 | 51.6 | 16.5 | 15.3 | 43.4 | 24.6 | 24.4 | 13.4 | 16.9 | 16.9 | 93.1 | 23.6 |
| WaterSearch | 13.5 | 51.6 | 20.9 | 17.4 | 50.5 | 36.0 | 28.6 | 50.5 | 20.1 | 25.3 | 91.4 | **31.4** |

Table 7: Comparison of WaterSearch and KGW-soft on Qwen-3-32B in 3K context length.

| Method | Short In/Short Out | | Short In/Long Out | | Long In/Short Out | | | Long In/Long Out | | Open-Ended | Overall | |
|---|---|---|---|---|---|---|---|---|---|---|---|---|
| | KoLA | Copen | ELI5 | FiQA | HotpotQA | LCC | RepoBench-P | Multinews | QMsum | AlpacaFarm | TP | GM |
| KGW-Soft | 9.3 | 43.4 | 11.4 | 11.1 | 34.5 | 14.0 | 14.0 | 9.7 | 15.5 | 4.0 | 97.4 | 16.7 |
| WaterSearch | 9.4 | 42.7 | 19.9 | 16.9 | 41.0 | 14.4 | 13.9 | 15.0 | 16.1 | 7.3 | 95.9 | **19.7** |

## D.3    COMPARISON ON LEXICAL-BASED AND SEMANTIC-BASED METHODS

This section compares the impact of lexical and semantic similarity metrics, using Rouge-L (Lin, 2004) and Sentence-BERT (Reimers & Gurevych, 2019), respectively. As shown in Table 8, Rouge-L achieves higher average performance overall, but the relative effectiveness varies across task types. For short-answer tasks such as KoLA and Copen, the two metrics yield comparable results. In

Table 8: Lexical-sequential and semantic similarity for chunk selection.

| Similarity Metric | KoLA | Copen | ELI5 | FiQA | HotpotQA | LCC | RepoBench-P | Multinews | QMsum | AlpacaFarm | Avg |
|---|---|---|---|---|---|---|---|---|---|---|---|
| Lexical | **2.4** | **27.2** | 12.4 | 16.0 | 16.3 | **27.6** | **27.1** | 13.6 | 13.5 | **4.5** | **16.1** |
| Semantic | 1.9 | **27.2** | **13.2** | **16.8** | **18.1** | 22.8 | 20.7 | **14.5** | **13.9** | 3.7 | 15.3 |

contrast, for comprehension-oriented tasks such as HotpotQA, MultiNews, and QMSum, Sentence-BERT performs better, as it captures semantic information in a vector space and evaluates similarity via cosine distance. For code completion tasks such as LCC and RepoBench-P, Sentence-BERT underperforms due to the lack of domain-specific training on source code, whereas longest common subsequence matching is more naturally aligned with the structural properties of code.

## E  DETECTION OF WATERSEARCH

While our sentence-level partitioning may seem vulnerable to targeted attacks (e.g., modifying sentence-final tokens), we emphasize that this vulnerability assumes the partitioning scheme is known to attackers. In practice, the partitioning strategy can be designed to remain agnostic to external observers, effectively mitigating such targeted attacks.

Table 9: Comparison on Insert Attack and Synonym Substitution Attack.

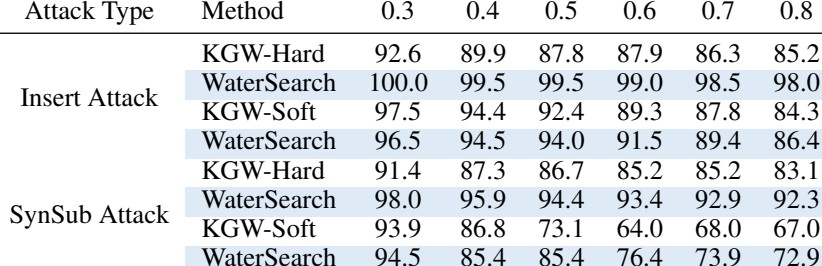

| Attack Type | Method | 0.3 | 0.4 | 0.5 | 0.6 | 0.7 | 0.8 |
|---|---|---|---|---|---|---|---|
| Insert Attack | KGW-Hard | 92.6 | 89.9 | 87.8 | 87.9 | 86.3 | 85.2 |
| | WaterSearch | 100.0 | 99.5 | 99.5 | 99.0 | 98.5 | 98.0 |
| | KGW-Soft | 97.5 | 94.4 | 92.4 | 89.3 | 87.8 | 84.3 |
| | WaterSearch | 96.5 | 94.5 | 94.0 | 91.5 | 89.4 | 86.4 |
| SynSub Attack | KGW-Hard | 91.4 | 87.3 | 86.7 | 85.2 | 85.2 | 83.1 |
| | WaterSearch | 98.0 | 95.9 | 94.4 | 93.4 | 92.9 | 92.3 |
| | KGW-Soft | 93.9 | 86.8 | 73.1 | 64.0 | 68.0 | 67.0 |
| | WaterSearch | 94.5 | 85.4 | 85.4 | 76.4 | 73.9 | 72.9 |

Table 10: Comparison on Paraphrase-Based Attacks.

| Method | Pegasus | Pegasus-Bigram | Parrot | Parrot-Bigram | AVG |
|---|---|---|---|---|---|
| KGW-Hard | 79.4 | 67.6 | 7.5 | 9.2 | 40.9 |
| WaterSearch | 90.2 | 74.2 | 20.6 | 19.6 | **51.2** |
| KGW-Soft | 79.4 | 67.6 | 10.3 | 9.2 | 41.6 |
| WaterSearch | 80.4 | 66.8 | 20.6 | 18.6 | **46.6** |

## F  ABLATION STUDY

### F.1  ABLATION ON PARALLEL NUMBER $K$

This part show the full experimental results of ablation on parallel number $K$. The experiment is conducted on Llama2-7B-Chat with $\gamma = 0.15, \delta = 10, \alpha = 0.75$. Text quality is judged by lexical and semantic similarity, respectively. Table 11 and Table 12 demonstrate the existance of Test time Scaling Law in both task completeness and detectability.

### F.2  ABLATION ON DIFFERENT PARALLEL METHODS

This section discuss the different in parallel searching paradigms under watermark settings, including beam search or rejection sampling. **For token-level constrained decoding** (like beam search), which purpose is maximize probability of a sequence, could fail to correct myopic distortions introduced at earlier tokens. **For post-hoc reranking** (repeat sampling under the same watermarking

Table 11: Scaling effect of WaterSearch in Generation and Detection by lexical similarity metric.

| Config | Short In/Short Out | | Short In/Long Out | | Long In/Short Out | | | Long In/Long Out | | Open-Ended | Overall | |
| --- | --- | --- | --- | --- | --- | --- | --- | --- | --- | --- | --- | --- |
| | KoLA | Copen | ELI5 | FiQA | HotpotQA | LCC | RepoBench-P | Multinews | QMsum | AlpacaFarm | TP | GM |
| K=2 | 1.9 | 27.1 | 13.5 | 15.7 | 17.0 | 24.1 | 22.6 | 13.5 | 13.5 | 2.8 | 94.1 | 15.2 |
| K=3 | 2.4 | 24.0 | 12.5 | 15.9 | 18.6 | 26.3 | 23.3 | 13.5 | 13.2 | 4.6 | 95.2 | 15.4 |
| K=4 | 2.4 | 27.2 | 12.4 | 16.0 | 16.3 | 27.6 | 27.1 | 13.6 | 13.5 | 4.5 | 97.6 | 16.1 |
| K=5 | 3.1 | 26.5 | 15.6 | 16.1 | 15.5 | 28.2 | 26.5 | 13.9 | 13.6 | 4.4 | 96.8 | 16.3 |
| K=6 | 3.0 | 27.3 | 12.5 | 16.4 | 17.8 | 29.4 | 27.1 | 13.7 | 13.5 | 4.3 | 96.9 | 16.5 |

Table 12: Scaling effect of WaterSearch in Genration and Detection by semantic similarity metric.

| Config | Short In/Short Out | | Short In/Long Out | | Long In/Short Out | | | Long In/Long Out | | Open-Ended | Overall | |
| --- | --- | --- | --- | --- | --- | --- | --- | --- | --- | --- | --- | --- |
| | KoLA | Copen | ELI5 | FiQA | HotpotQA | LCC | RepoBench-P | Multinews | QMsum | AlpacaFarm | TP | GM |
| K=2 | 1.9 | 24.5 | 13.4 | 16.6 | 15.1 | 23.0 | 24.1 | 14.2 | 13.3 | 4.7 | 93.9 | 15.1 |
| K=3 | 2.3 | 24.5 | 12.7 | 16.2 | 17.4 | 21.9 | 21.3 | 14.8 | 13.6 | 4.7 | 95.7 | 14.9 |
| K=4 | 1.9 | 27.2 | 13.2 | 16.8 | 18.1 | 22.8 | 20.7 | 14.5 | 13.9 | 4.7 | 97.9 | 15.4 |
| K=5 | 3.0 | 27.1 | 12.4 | 17.0 | 16.1 | 23.3 | 23.6 | 14.3 | 13.9 | 4.2 | 97.0 | 15.5 |
| K=6 | 2.8 | 32.0 | 13.0 | 16.7 | 15.7 | 24.6 | 22.7 | 14.9 | 13.5 | 4.7 | 97.0 | 16.1 |

Table 13: Ablations on different parallel searching methods (Beam number $B = 5$).

| Method | Short In/Long Out | | Short In/Long Out | | Long In/Short Out | | | Long In/Long Out | | Open-Ended | Overall |
| --- | --- | --- | --- | --- | --- | --- | --- | --- | --- | --- | --- |
| | KoLA | Copen | ELI5 | FiQA | HotpotQA | LCC | RepoBench-P | Multinews | QMsum | AlpacaFarm | GM |
| Beam Search | 1.0 | 7.6 | 4.4 | 8.7 | 3.0 | 17.4 | 16.7 | 6.9 | 10.1 | 3.4 | 7.9 |
| KGW-Soft (post-hoc) | **2.2** | 24.3 | 12.4 | 14.7 | **20.1** | 26.3 | 26.6 | 12.3 | **13.9** | **4.2** | 15.7 |
| WaterSearch | 1.91 | **27.2** | **13.2** | **16.8** | 18.1 | **27.6** | **27.1** | **14.5** | 13.9 | 3.7 | **16.4** |

scheme), it could increase the diversity of generation attributed to sampling probability, but it cannot induce more diverse variations caused by watermark priors. For example, the green list partition prior can mostly determine the color of a token, which is a common problem in short text or low entropy scenarios. **For WaterSearch**, it explicitly increases candidate diversity through multiple random seeds, each producing a different watermark perturbation (like partition of vocabulary in KGW). Moreover, WaterSearch dynamically updates the prefix after each selected chunk, which helps mitigate semantic drift compared to post-hoc approaches that rely on a fixed reference continuation.

Table 13 compares WaterSearch to beam search and rejection sampling under the 5 beams of search to illustrate this difference. WaterSearch gets highest score among all. Beam search performs worst, showcase the token-level perturbation is less effective than others. For short text generation, post-hoc method gains similar scores as ours. While in long output tasks like ELI5, FiQA and Multinews, our method has a 0.8, 2.1, 2.2 improvement, which further verifies our assumption about semantic drift.

## F.3 ABLATION ON FACTORS $\alpha$ TO BALANCE TEXT QUALITY AND DETECTABILITY

In this section, we present a comprehensive analysis of how the coefficient $\alpha$ influences both downstream task performance and watermark detectability. Using KGW-soft as our baseline watermarking method, we conduct evaluations on 4 datasets, Copen (Peng et al., 2022),ELI5 (Fan et al., 2019),HotpotQA (Yang et al., 2018) and MultiNews (Fabbri et al., 2019), on Llama-2-7B-chat to systematically assess these effects. Our experiment is designed in low / high watermarking strength and ROUGE-L / semantic similarity settings. Hyper-parameters are defined as: $\alpha = [0, 0.25, 0.5, 0.75, 1]$, for low strength $\gamma = 0.15, \delta = 5.0$ and for high strength $\gamma = 0.15, \delta = 10.0$.

The experiment result is illustrated from Figure 6 to 9. We find that as $\alpha$ increases, the watermark strength enhances, while the text quality degrades within a controllable range. Besides, the resulting Pareto-optimal curve consistently lies above that of the baseline methods, and the advantage becomes even more pronounced in regimes that require higher watermark strength. This indicates WaterSearch's effectiveness and robustness.

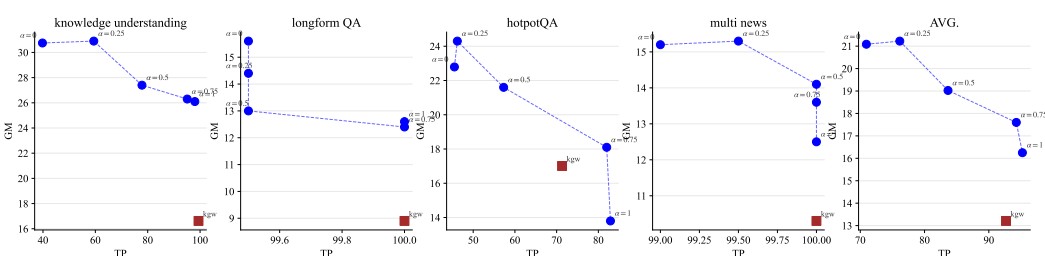

Figure 6: Trade-off between text quality and detectability in high watermarking strength and Rouge-L metric settings.

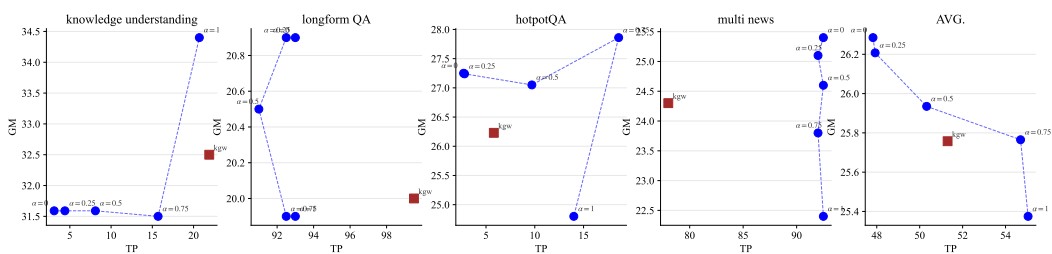

Figure 7: Trade-off between text quality and detectability in low watermarking strength and Rouge-L metric settings.

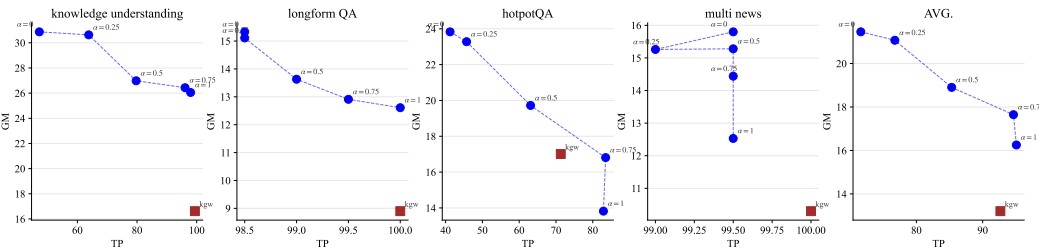

Figure 8: Trade-off between text quality and detectability in high watermarking strength and semantic similarity metric settings.

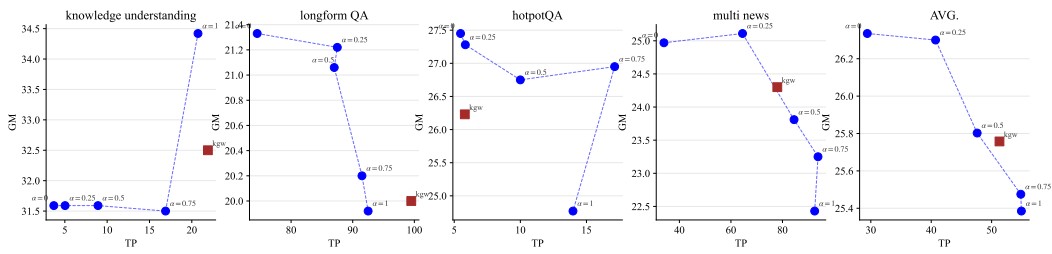

Figure 9: Trade-off between text quality and detectability in low watermarking strength and semantic similarity metric settings.

### F.4 ABLATION ON CHUNK SIZE

This part discuss the ablation on chunk size (token number) of long output tasks. Here we set $\gamma = 0.15, \delta = 10, \alpha = 0.75$. The result is shown in Table 14, and for long output tasks (512 tokens maximum), increase chunk size could improve the performance to some extent by aliviating semantic drift as time.

Table 14: Ablation on chunk size (tokens) of long output tasks.

| Task | Metric | 20 | 40 | 60 | 80 |
|------|--------|------|------|------|------|
| multi_news | GM | 13.62 | 13.84 | 13.71 | 13.92 |
|  | TP | 100 | 100 | 100 | 100 |
| qmsum | GM | 13.49 | 13.59 | 13.63 | 13.54 |
|  | TP | 100 | 100 | 100 | 100 |

## G CASE STUDY

We choose examples from evaluation logs in this section. Listing 1 and 2 demonstrate the dynamic chunk selection process. They ahere to the meaning of the standard text and choose a correct answer at last. Listing 3, 4, 5 and 6 show the generated text. WaterSearch can generate fluent and meaningful sentences compared to the repetition of their counterpart.

```
Input:
Please give answer to the following question about knowledge. Note: If
    you are asked for true or false, just answer "true" or "false" only.
    If you are asked for similarity, just answer with the entity name
    only. Do not give anything other than the answers.
Question:
Among TRAPPIST-1g, nandrolone, BL 6 inch Mk VII naval gun, Kanchenjunga,
    1923 United Kingdom general election, 8.8 cm SK L/45 naval gun,
    police auction, Downtown Hudson Tubes, Lebanon, 1982 FIFA World Cup,
    Oswald Poche, Wheat leaf rust, Via Lattea, Fortress of Mimoyecques,
    Battle of Jutland, Ford Pinto engine, Xynotyro, rs6314, 2008 Sichuan
    earthquake, Solar eclipse of July 16, 2186, Parrott rifle, which one
    is the most conceptually similar with Belgium?
Answer:
Lebanon
Vanilla kgw:
BL
Watersearch:
Standard text generated: Lebanon
Candidate 1: Lebanon
Candidate 2: Kanjenchhana
Candidate 3: lebonostregeography
Candidate 4: Lebanon
Choose candidate 4
```

Listing 1: A WaterSearch Example in HotpotQA

```
Input:
The following are given passages.
Passage 1:
Primary Care Behavioral health
The primary care behavioral health (PCBH) consultation model is a
    psychological approach to ...
Answer the question based on the given passages. Only give me the answer
    and do not output any other words.
Question: Who did Fredric Rieders tesify agains who had killed as many as
    60 patients and recieved three consecutive life terms to be served
    in Florence Colorado?
Answer:
```

```
Michael Swango
Vanilla kgw:
Daniel J. Swago
Watersearch:
Standard text generatedMichael Swango
Candidate 1: Fredric Rieders testified against Michael Joseph Swango, who
    was found guilty of murdering approximately as many as ...
Candidate 2: Michael Swango
Candidate 3: Fredric Rieders testified against Michael Swango, who it's
    believed could be responsible for as many as ...
Candidate 4: Michael Swango
Choose candidate 2
```

Listing 2: A WaterSearch Example in Copen

```
Question:
You are a helpful assistant, please answer the following question within
    300 words:
Wind ? Where the hell does it originate from and how does it work ?
    Something that has blown right past me all my life . I kinda get the
    concept , but where does wind begin ? Explain like I'm five.
Answer:
There 's one concept you have to understand first : air pressure .
    Basically , it 's how dense the air is in a particular area . This is
     affected by what 's in the air ...
Vanilla kgw:
Awesome! *cravendroneems* Well ~ the wind *fancytwirl* actually starts
    way *fancytwirlbigtwirlbigwinkleseaweeinspace* out there. Like
    reallyway way *fancytwirlbigtwirlbigwinkleseaweeinspace* out
    ththereee looks *fancytwirlbigtwirlbigwinkleseaweeinspace* a tiny bit
     messed ~ in *fancytwirlbigtwirlbigwinkleseaweeinspace* a tiny bit
    messed ~ in
Watersearch:
Wind originate? How amazing, that is such  excising question I have a
    feeling most people don not know even a little, but here let ' I
    explain it  very  patient ly :  The concept where something is going
    away might not come natural to everyone because of course wind isn
    not a human being but rather a type force and its a type force
    because it can be measured and proven scientificly. That means wind,
    as in gase, ous particles within a medium that moves past or transfer
     energy thrgh it and is also another thing ...
```

Listing 3: Case Study 1 from Evaluation

```
Question:
You are a helpful assistant, please answer the following question with
    financial knowledge within 300 words:
What is the point of the stock market? What is it for, and why might
    someone want to trade or invest?
Answer:
In finance, form is function, and while a reason for a trade could be
    anything, but since the result of a trade is a change in value, it
    could be presumed that one seeks to receive a change in value
Vanilla kgw:
Sure thing! Thestock exchange is essentiallya  venuforgktmembersto
    purchaseandexcbstocertain businessestooftypeanystacksof their
    securities, knownasa"ownunks". Thesessionmkaibtoprovemsters
    aplatformor Wheredonmves theprice oft  a business'security.
    Thesessionmkaibtoprovemsters  aplatformor Wheredonmves theprice oft
    a business'security. Thesessionmkaibtoprovemsters  aplatformor
    Wheredonmves theprice oft  a business'security.
    Thesessionmkaibtoprovemsters  aplatformor Wheredonmves theprice oft
    a business'security.
Watersearch:
```

```
Sure, here are  some answers up top! If those fail I'll do better with
    greater clarity:  the stock market brings supply of goods or stock is
     sold giving the stock market entity a chance to get their stock onto
     shoppings radars and receive investers funds and if customers like
    items they'll have an easier to purchase it in this manner and get
    more money for each item as well.
The historical point or historical use for this is farmer Ben and a bunch
    on fares in Philadelphia ...
```

Listing 4: Case Study 2 from Evaluation

```
Question:
Answer the question based on the given passages. Only give me the answer
    and do not output any other words.
The following are given passages.
Passage 1:
English general election, 1689
The 1689 English general election, held in January 1689, elected the
    Convention Parliament, which was summoned in the aftermath of the
    Glorious  R e v o l u t i o n
Answer the question based on the given passages. Only give me the answer
    and do not output any other words.\
Question: Who was the last monarch of England to be overthrown before The
     English General Election, 1690?
Answer:
King James II of England
Vanilla kgw:
The  Glorius Revolution of 1689 resulted in Charles II, the previous last
     King before this, being overborne/removed as he gave
Watersearch:
King James II
```

Listing 5: Case Study 3 from Evaluation

```
Question:
Please give answer to the following question about knowledge. Note: If
    you are asked for true or false, just answer "true" or "false" only.
    If you are asked for similarity, just answer with the entity name
    only. Do not give anything other than the answers. Question:
Among Mall of America, adenosine diphosphate, Trawniki concentration camp
    , Ignalina Nuclear Power Plant, Ford Windsor engine, Lake Winnipeg,
    Bronx Zoo, Gliese 581g, Three Mile Island Nuclear Generating Station,
     8.8 cm SK L/30 naval gun, Katiki Domokou, Australia, Battle of
    Chancellorsville, daf-5, Pontiac V8 engine, BMW M54, The Four Seasons
     Restaurant, Travelodge, Crawford Notch, CR2032 battery, Buick V8
    engine, which one is the most conceptually similar with M61 Vulcan?
Answer:
8.8 cm SK L/30 naval gun
Vanilla kgw:
eightinch_fortr02_mlx85-gun
Watersearch:
8.8 cm SK L/30 naval gun
```

Listing 6: Case Study 4 from Evaluation

## H  DECLARATION OF LARGE LANGUAGE MODEL UTILIZATION

In accordance with the ICLR policy on the use of large language models (LLMs), we hereby declare that LLMs were employed solely to assist in improving the grammar and enhancing the expression of this paper. The original research idea, methodological development, and overall structure and content of the manuscript were entirely conceived and written by the authors. At no stage was the use of LLMs extended to the generation of core intellectual content, and we affirm that there has been no misuse of LLMs in the preparation of this work.

