# OpenReview forum: "WaterSearch: A Quality-Aware Search-based Watermarking Framework for Large Language Models"
_ICLR.cc/2026/Conference — ICLR 2026 Conference Withdrawn Submission_

### Official Review · Reviewer_Qo61 · 2025-10-31

**Soundness:** 3
**Presentation:** 3
**Contribution:** 2
**Rating:** 4
**Confidence:** 4

**Summary:**

By constraining semantic consistency with the original sentence as much as possible, generating multiple outputs with different random seeds, and combining these outputs, this paper identified the optimal balance between semantic integrity and watermarking.

**Strengths:**

1. The design of alpha is rigorous overall, with theoretical proof of its effectiveness and ablation studies demonstrating optimal alpha values, reasonably extending the KGW method.
2. Effectively designed time complexity to ensure computational resources increase only moderately.
3. Experiments demonstrate that sufficiently large differences between random seeds enable multiple outputs of the watermark to combine into text with semantics closer to the original meaning, including the validity of other hyperparameters such as K.

**Weaknesses:**

1. Missing Visualization examples of all results, just an NBA example
2. Scoring q is a linear add-up of semantic similarity towards the original output, and watermarking quality, which is very straight forward, but can be questioned that if the linear add-up is effective or not, more theoritical supports are needed
3. Strategy of picking different random seed is still not clear enough for me

**Questions:**

How will the method perform on bigger and SOTA language models?

---

> ### Author Response · Authors · 2025-11-29
> **Response to Reviewer Qo61 (1/4)**
>
> We thank the reviewer for the valuable feedback on improving this paper! Please find below our response to the reviewer’s questions.
>
> > Q1: How will the method perform on bigger and SOTA language models?
>
> Thank you for pointing that out, we have tested WaterSearch on Qwen-3-14B and Qwen-3-32B to fully assess our method as below, limited by computing source, we control model's context length to 4K (14B) and 3K (32B):
> |  Method       | KoLA  | Copen | ELI5  | FiQA  | HotpotQA | LCC   | RepoBench-P | Multinews | QMsum  | AlpacaFarm | AVG   |
> |--------------|-------|-------|-------|-------|----------|-------|-------------|-----------|--------|------------|-------|
> |  KGW-Soft     | 12.6  | 51.6  | 16.5  | 15.3  | 43.4     | 24.6  | 24.4        | 13.4      | 16.9   | 16.9       | 23.6  |
> | WaterSearch  | 13.5  | 51.6  | 20.9  | 17.4  | 50.5     | 36.0  | 28.6        | 50.5      | 20.1   | 25.3       | **31.4**  |
>
> | Method       | KoLA  | Copen | ELI5  | FiQA  | HotpotQA | LCC   | RepoBench-P | Multinews | QMsum  | AlpacaFarm | AVG   |
> |--------------|-------|-------|-------|-------|----------|-------|-------------|-----------|--------|------------|-------|
> | KGW-Soft     | 9.3   | 43.4  | 11.4  | 11.1  | 34.5     | 14.0  | 14.0        | 9.7       | 15.5   | 4.0        | 16.7  |
> | WaterSearch  | 9.4   | 42.7  | 19.9  | 16.9  | 41.0     | 14.4  | 13.9        | 15.0      | 16.1   | 7.3        | **19.7** |
>
> As shown in the 2 tables above, our method could outperform the baseline method in bigger and SOTA LLMs, which further showcase the effectiveness of our method.

---

> ### Author Response · Authors · 2025-11-29
> **Response to Reviewer Qo61 (2/4)**
>
> > Q2: Missing Visualization examples of all results
>
> Here are 2 visualization examples from Copen and HotpotQA,
>
> ```
> Input:
> Please give answer to the following question about knowledge. Note: If you are asked for true or false, just answer "true" or "false" only. If you are asked for similarity, just answer with the entity name only. Do not give anything other than the answers. Question:
> Among TRAPPIST-1g, nandrolone, BL 6 inch Mk VII naval gun, Kanchenjunga, 1923 United Kingdom general election, 8.8 cm SK L/45 naval gun, police auction, Downtown Hudson Tubes, Lebanon, 1982 FIFA World Cup, Oswald Poche, Wheat leaf rust, Via Lattea, Fortress of Mimoyecques, Battle of Jutland, Ford Pinto engine, Xynotyro, rs6314, 2008 Sichuan earthquake, Solar eclipse of July 16, 2186, Parrott rifle, which one is the most conceptually similar with Belgium?
>
> Answer:
> Lebanon
>
> Vanilla kgw:
> BL
>
> Watersearch:
> Standard text generated：Lebanon
> Candidate 1: Lebanon
> Candidate 2: Kanjenchhana
> Candidate 3: lebonostregeography
> Candidate 4: Lebanon
>
> Choose candidate 4
> ```
>
> ```
> Input:
> The following are given passages.
> Passage 1:
> Primary Care Behavioral health
> The primary care behavioral health (PCBH) consultation model is a psychological approach to ...
> Answer the question based on the given passages. Only give me the answer and do not output any other words.
> Question: Who did Fredric Rieders tesify agains who had killed as many as 60 patients and recieved three consecutive life terms to be served in Florence Colorado?
>
> Answer:
> Michael Swango
>
> Vanilla kgw:
> Daniel J. Swago
>
> Watersearch:
> Standard text generated：Michael Swango
> Candidate 1: Fredric Rieders testified against Michael Joseph Swango, who was found guilty of murdering approximately as many as ...
> Candidate 2: Michael Swango
> Candidate 3: Fredric Rieders testified against Michael Swango, who it's believed could be responsible for as many as ...
> Candidate 4: Michael Swango
>
> Choose candidate 2
> ```
> More visualization examples could be found in Appendix G for better understanding. We really appreciate the reviewer's suggestion.

---

> ### Author Response · Authors · 2025-11-29
> **Response to Reviewer Qo61 (3/4)**
>
> > Q3: Strategy of picking different random seed is still not clear enough
>
> Unlike previous methods that use a secret key to manipulate distribution, we design a novel scheme to control a seed pool to enable diverse parallel generation. The process can be formalized as
>
> $Seed\\_Pool = Hash([s_1,s_2,\cdots],key)=shuffle([s_1,s_2,\cdots],key)$
>
> Where $[s_1,s_2,\cdots]$ is a large group of seeds, and the hash function could be shuffle condition of key.
> Besides, the token-level seed generation in the KGW framework can be expressed as:
>
> $Seed = Hash(x^{-h},\cdots,x^{-2},x^{-1})=(x^{-h}\cdot x^{-2}\cdot x^{-1}) \mod M$
>
> Where $seed$ is retrieved from a seed pool and $M$ is a large prime number. This mechanism  is widely discussed in [1] and here we set $h=1$ and align with KGW's settings.
> Below, we demonstrate the implementaion in simplified pseudocode:
>
> ```
> ## Generation
> random.seed(41) # pre-defined scret key
> self.seed_pool = list(range(1, 5000)) # initialize seed pool
> for _ in range(L//m): # iteration over each chunk
>     random.shuffle(self.seed_pool) # build seed pool
>     cur_seeds = self.seed_pool[:self.K]
>     no_wm_chunk = model.generate(input_text_ids)
>     wm_chunks = []
>     for i in range(K):
>     # iteraction over each beam for search
>     # simplify for demonstration, achieve in parallel actually
>         token_seed = hash_func(input_text_ids[-h:], cur_seeds[i]) # build token-level seed
>         wm_chunk = model.generate(input_text_ids, wm_logit_processor, token_seed)
>         wm_chunks.append(wm_chunk)
>     idx = calcuate_best_wm_chunk(no_wm_chunk, wm_chunks) # calculate the best wm_chunk
>     input_text_ids = concat(input_text_ids, wm_chunks[idx])
>
> return tokenizer.decode(input_text_ids)
> ```
> ```
> ## Detection
> random.seed(41) # pre-defined scret key
> self.seed_pool = list(range(1, 5000)) # initialize seed pool
> chunk_wm_scores = []
> for i in range(L//m): # iteration over each chunk
>     random.shuffle(self.seed_pool) # build the same seed pool
>     cur_seeds = self.seed_pool[:self.K] # recover seeds in generation
>     chunk_text = output_text[i*m: (i+1)*m]
>     chunk_wm_score = chunk_detect(chunk_text, cur_seeds) # chunk-level detection
>     chunk_wm_scores.append(chunk_wm_score)
>
> p_value = document_detect(chunk_wm_scores) # document-level detection by chi^2 statistic test
> return p_value
> ```
> It is worth notice that WaterSearch employs a $\chi^2$ statistic test to evaluate the probability of each chunk to return confidence overly, which makes token-level perturbation hard to influence detection accuracy. The experimental result against different types of attack is shown in Section 6.3. More discussion of seed generation mechanism is included in Appendix A for better understanding (highlighted in darkred). We really appreciate the reviewer's suggestion.
>
> [1] Kirchenbauer et al., A watermark for large language models. ICML 2023.

---

> ### Author Response · Authors · 2025-11-29
> **Response to Reviewer Qo61 (4/4)**
>
> > Q4: Scoring q is a linear add-up of semantic similarity towards the original output, and watermarking quality, which is very straight forward, but can be questioned that if the linear add-up is effective or not, more theoretical supports are needed
>
> In the main results, we did not heavily tune the value of $\alpha$ but fixed it to 0.75 in most settings, which already provides great performance. We thank the reviewer for noticing this point. In response, we conducted extensive experiments to examine the effect of $\alpha$. In detail, we design scenarios of low / high watermarking strength and lexical / semantic and choose 4 representative datasets for experiment. The results overview is shown in the table below:
>
> | Type                | Metric | 0 | 0.25 | 0.5 | 0.75 | 1 |
> |---------------------|--------|------------|---------------|--------------|---------------|------------|
> | lexical, high strength | GM     | 13.2       | 21.1          | 21.2         | 19.0          | 17.6       |
> |                     | TP     | 92.7       | 71.0          | 76.2         | 83.7          | 94.3       |
> | lexical, low strength  | GM     | 25.8       | 26.3          | 26.2         | 25.9          | 25.8       |
> |                     | TP     | 51.3       | 47.8          | 47.9         | 50.3          | 54.7       |
> | semantic, high strength | GM    | 13.2       | 21.5          | 21.1         | 18.9          | 17.6       |
> |                     | TP     | 92.7       | 71.6          | 76.8         | 85.3          | 94.7       |
> | semantic, low strength  | GM    | 25.8       | 26.3          | 26.3         | 25.8          | 25.5       |
> |                     | TP     | 51.3       | 29.4          | 40.7         | 47.6          | 54.8       |
>
> We observe that increasing $\alpha$ strengthens the watermark signal while causing a controlled degradation in text quality, indicating robustness to the choice of $\alpha$. Besides, the resulting Pareto-optimal curve consistently lies above that of the baseline methods, and the advantage becomes even more pronounced in regimes that require higher watermark strength.
> Please find the visualized experimental results and discussion in Appendix F.3.
>
> ---
> Again, we thank the reviewer for the valuable feedback. Please let us know if there are any other questions or suggestions.
>
> Best,
>
> Authors

---

### Official Review · Reviewer_V6X7 · 2025-11-01

**Soundness:** 3
**Presentation:** 4
**Contribution:** 2
**Rating:** 4
**Confidence:** 4

**Summary:**

This paper introduces WaterSearch, a search-based watermarking framework that aims to improve the trade-off between watermark detectability and text quality in large language models (LLMs). Instead of modifying logits during generation as in the standard KGW framework, WaterSearch performs chunk-level parallel generation: it generates multiple candidate continuations (some watermarked, one unwatermarked) and selects the one that maximizes a joint score balancing semantic similarity to the unwatermarked text and detectability based on green-list token frequency. A chi-square–based detection procedure is proposed to test statistical significance across chunks. Experiments across 3 LLMs (Llama-2, Qwen-2.5, InternLM) and 10 datasets show consistent improvements in both generation quality and detection robustness, particularly under low-entropy and short-text conditions.

**Strengths:**

* Simple idea and framework: WaterSearch can be applied on top of existing KGW-style watermarking schemes with minimal modification.
* The method improves performance across all evaluated datasets, including difficult cases such as short-text or low-entropy settings, where KGW tends to fail.
* The paper uses WaterBench and additional benchmarks (e.g., RepoBench-P) and shows gains across multiple model families.
* Figures and algorithm descriptions make the approach easy to follow; the writing is concise and readable.

**Weaknesses:**

* Incremental conceptual novelty: The idea to generate several watermarked candidates and pick the best is intuitive, but very closely resembles beam search or rejection sampling. The contribution feels more engineering-oriented than conceptual, especially given that most of the theoretical development restates expected properties of the existing KGW trade-off.

* Computational inefficiency: WaterSearch performs parallel or beam-style generation of multiple watermarked candidates per chunk and selects the best one, which intuitively incurs substantial wall-time cost. While Table 4 discusses space complexity, runtime overhead or throughput (tokens/s) is not reported. Without this, it is hard to assess practical efficiency, but based on the runtime complexity reported in the paper, a ~5x slowdown in generation is fairly substantial and reduces the practical utility of the method.

**Questions:**

* What is the actual computational overhead relative to vanilla KGW? Reporting wall-clock time or tokens/s for each configuration would clarify practical feasibility.

* How sensitive is the approach to the number of parallel candidates k? Does increasing k yield linear improvement in detectability, or diminishing returns?

* Could the same results be achieved by post-hoc reranking or constrained decoding (e.g., using logits rather than full re-generation)?

* Since quality is evaluated only relative to the unwatermarked model’s continuation, could semantic drift still occur if that baseline itself is low-quality or inconsistent?

* The claim that KGW “maintains text quality well from the perspective of perplexity or LLM-as-judge” is contradicted by results in WaterBench (Tu et al., 2024) and New Evaluation Metrics Capture Quality Degradation due to LLM Watermarking (Singh et al., 2024), which show measurable degradations in both perplexity and subjective fluency for KGW. The discussion should acknowledge these findings.

---

> ### Author Response · Authors · 2025-11-29
> **Response to Reviewer V6X7 (1/4)**
>
> We thank the reviewer for the valuable feedback on improving this paper! Please find below our response to the reviewer’s questions.
>
> > Q1: What is the actual computational overhead relative to vanilla KGW? Reporting wall-clock time or tokens/s for each configuration would clarify practical feasibility.
>
> Thank you for this very practical question. We report throughput and peak memory consumption across four datasets: Copen (short input / short output), ELI5 (short input / long output), HotpotQA (long input / short output), and MultiNews (long input / long output). All experiments are conducted on a single NVIDIA A800 GPU.
>
> | Generation Throughput | KGW  | Beam=2 | Beam=3 | Beam=4 | Beam=5 | Beam=6 |
> |-----------------------|------|--------|--------|--------|--------|--------|
> | Copen                 | 55.8 | 44.9   | 41.3   | 42.1   | 39.1   | 39.6   |
> | ELI5                  | 57.0 | 47.3   | 46.3   | 45.4   | 45.4   | 46.0   |
> | HotpotQA              | 23.3 | 11.6   | 8.3    | 5.3    | 4.1    | 3.9    |
> | MultiNews             | 54.7 | 46.5   | 34.9   | 31.1   | 29.8   | 23.6   |
>
> | Peak Memory Usage | KGW  | Beam=2 | Beam=3 | Beam=4 | Beam=5 | Beam=6 |
> |-------------------|------|--------|--------|--------|--------|--------|
> | Copen             | 12.6 | 12.8   | 12.9   | 13.0   | 13.1   | 13.2   |
> | ELI5              | 12.8 | 13.6   | 14.0   | 14.5   | 14.8   | 15.3   |
> | HotpotQA          | 14.9 | 21.2   | 24.7   | 28.1   | 31.5   | 35.0   |
> | MultiNews         | 14.9 | 21.8   | 25.8   | 30.2   | 32.6   | 36.7   |
>
> WaterSearch introduces additional cost, which is expected given its design. However, on short-input tasks such as Copen and ELI5, the impact on throughput is minimal and the increase in peak memory is modest. For long-input tasks, including HotpotQA and MultiNews, the overhead becomes more apparent; nonetheless, the growth remains sub-linear, demonstrating that WaterSearch scales efficiently in practice. Importantly, the method achieves substantial quality gains even at small parallel degrees (e.g., $k=2$, see Section 6.4), providing flexible hyper-parameters that can be tuned to accommodate different computational budgets. Please find the visualization, more results and discussion in Section 6.5 (highlighted in darkred).

---

> ### Author Response · Authors · 2025-11-29
> **Response to Reviewer V6X7 (2/4)**
>
> > Q2: How sensitive is the approach to the number of parallel candidates k? Does increasing k yield linear improvement in detectability, or diminishing returns?
>
> Thanks for this very practical question. We conduct experiments under two settings, lexical versus semantic similarity, and report results for text quality (GM) and detection precision (TP) as below:
>
> | similarity | metric | KGW-Soft | K=2  | K=3  | K=4  | K=5  | K=6  |
> |------------|--------|----------|------|------|------|------|------|
> | lexical    | GM     | 11.8     | 15.2 | 15.4 | 16.1 | 16.3 | 16.5 |
> |            | TP     | 95.3     | 94.1 | 95.2 | 97.6 | 96.8 | 96.9 |
> | semantic   | GM     | 11.8     | 15.1 | 14.9 | 15.4 | 15.5 | 16.1 |
> |            | TP     | 95.3     | 93.9 | 95.7 | 97.9 | 97.0 | 97.9 |
>
> The result demonstrates the scaling effect of both GM and TP with the increase of $K$. Notably, our method outperforms the baseline ($GM=11.8$) even when $K=2$, and as $K$ increases, the detection accuracy gradually exceeds that of the baseline ($TP=95.3$). Please find the visualization and full result in Section 6.4 And Appendix F.1 (highlighted in darkred).

---

> ### Author Response · Authors · 2025-11-29
> **Response to Reviewer V6X7 (3/4)**
>
> > The idea to generate several watermarked candidates and pick the best is intuitive, but very closely resembles beam search or rejection sampling.
>
> Thank you for pointing out the commonalities among different methods. We would like to first clarify the we first notice watermark embedding is determined by random seeds. Building on this insight, we design a novel watermark embedding scheme that manipulates a seed pool rather than individual seeds during each token generation. The findings of accumulative distribution difference in seeds, as illustrated in Figure 2 further enables parallelized text generation and detection. Based on that, we could design a parallel searching framework. Please find the pseudocode of seed generation process and the mechanism in appendix A for better understanding.
>
> > Q3: Could the same results be achieved by post-hoc reranking or constrained decoding (e.g., using logits rather than full re-generation)?
>
> We appreciate your practical concern. We would like to introduce the main difference between beam search or rejection sampling and our method.
>
> **For token-level constrained decoding** (like beam search), they could fail to correct myopic distortions introduced at earlier tokens.
>
> **For post-hoc reranking** (repeat sampling under the same watermarking scheme), it could increase the diversity of generation attributed to sampling probability, but it cannot induce more diverse variations caused by watermark priors. For example, the green list partition prior can mostly determine the color of a token, which is a common problem in short text or low entropy scenarios.
>
> **For WaterSearch**, it explicitly increases candidate diversity through multiple random seeds, each producing a different watermark perturbation (like partition of vocabulary in KGW). Moreover, WaterSearch dynamically updates the prefix after each selected chunk, which helps mitigate semantic drift compared to post-hoc approaches that rely on a fixed reference continuation.
> To illustrate this difference in practice, we compared WaterSearch to beam search and rejection sampling under the 5 beams of search.
>
> | K=5            | KoLA  | Copen | ELI5  | FiQA  | HotpotQA | LCC   | RepoBench-P | Multinews | QMsum  | AlpacaFarm | AVG   |
> |----------------|-------|-------|-------|-------|----------|-------|-------------|-----------|--------|------------|-------|
> | Beam Search    | 1.0   | 7.6   | 4.4   | 8.7   | 3.0      | 17.4  | 16.7        | 6.9       | 10.1   | 3.4        | 7.9   |
> | Rejection Sampling | 2.2 | 24.3 | 12.4 | 14.7 | 20.1    | 26.3  | 26.6        | 12.3      | 13.9   | 4.2        | 15.7  |
> | WaterSearch    | 1.91  | 27.2  | 13.2  | 16.8  | 18.1     | 27.6  | 27.1        | 14.5      | 13.9   | 3.7        | 16.4  |
>
> Beam search performs worst among all, showcase the token-level perturbation is less effective than others. For short text generation, post-hoc method gains similar scores as ours. While in long output tasks like ELI5, FiQA and Multinews, our method has a 0.8, 2.1, 2.2 improvement, which further verifies our assumption about semantic drift.  Detailed experimental result could be found in Appendix F.2.

---

> ### Author Response · Authors · 2025-11-29
> **Response to Reviewer V6X7 (4/4)**
>
> > Q4: Since quality is evaluated only relative to the unwatermarked model’s continuation, could semantic drift still occur if that baseline itself is low-quality or inconsistent?
>
> 1. The unwatermarked continuation is not treated as a globally optimal solution, but rather as a way to preserve the model’s inherent distributional behavior. It provides a reference trajectory that reflects how the base model intends to continue the text.
> 2. Empirically, semantic drift is substantially reduced compared to token-level watermarking, because WaterSearch evaluates full-chunk semantic consistency instead of making token-level decisions. This effect is consistently observed across our experiments (e.g., Table 1), where WaterSearch preserves meaning more faithfully while achieving stronger detectability.
> 3. It is true that the reference continuation is conditioned on previously selected watermarked chunks, and thus semantic drift could in principle accumulate over time. However, in practice:
>  a) The first vanilla chunk serves as a strong semantic anchor, grounding the generation in the model’s original distribution and largely determining the overall intent and topic of the text.
>  b) The chunk length can be adapted to different tasks, and for long-form generation we specifically conduct an ablation study on the chunk size as below:
>
> | Chunk Size | MultiNews       |        | QMSum           |        |
> |------------|-----------------|--------|-----------------|--------|
> |                    | TP              | GM     | TP              | GM     |
> | 20         | 100             | 13.62  | 100             | 13.49  |
> | 40         | 100             | 13.84  | 100             | 13.59  |
> | 60         | 100             | 13.71  | 100             | 13.63  |
> | 80         | 100             | 13.92  | 100             | 13.54  |
>
> Here we set $\gamma=0.15,\delta=10,\alpha=0.75$. The result demonstrates that experimental performance shows an increasing trend with the growth of text chunk length, which can alleviate semantic drift to a certain extent.
>
> ---
> > Q5: The claim that KGW “maintains text quality well from the perspective of perplexity or LLM-as-judge” is contradicted by results in WaterBench (Tu et al., 2024) and New Evaluation Metrics Capture Quality Degradation due to LLM Watermarking (Singh et al., 2024), which show measurable degradations in both perplexity and subjective fluency for KGW. The discussion should acknowledge these findings.
>
> While perplexity used in KGW [1] could capture basic coherence, it is a relatively weak measure of text quality. LLM-as-judge [2] evaluates multiple dimensions such as coherence, clarity, and relevance, revealing finer-grained degradations. This work emphasizes task-oriented quality: generated text should preserve downstream task performance. WaterBench [3] shows that existing watermarking methods degrade drastically, which motivates this work to tackle the problem. We have updated the discussion on our paper. Please find the comparison in Section 2.2 (highlighted in darkred).
>
> [1] Kirchenbauer et al., A watermark for large language models. ICML 2023
>
> [2] Singh et al., New Evaluation Metrics Capture Quality Degradation due to LLM Watermarking. TMLR 2024
>
> [3] Tu et al., WaterBench: Towards holistic evaluation of watermarks for large language models. ACL 2024
>
> ---
> Again, we thank the reviewer for the valuable feedback. Please let us know if there are any other questions or suggestions.
>
> Best,
>
> Authors

---

### Official Review · Reviewer_xzV6 · 2025-11-01

**Soundness:** 3
**Presentation:** 3
**Contribution:** 2
**Rating:** 2
**Confidence:** 4

**Summary:**

The paper introduces WaterSearch, a search-based LLM watermarking framework that generates multiple parallel candidates instead of a single one and then selects the one that best preserves coherence with the original text to enhance quality and detectability.

**Strengths:**

1. The idea of using multiple candidates is clear and effective.

2. The paper also includes a solid theoretical analysis of the proposed method.

3. Evaluations are comprehensive, on various models and tasks.

**Weaknesses:**

1. The overhead of this method seems to be significant.

2. A main weakness of the paper is the limited comparison against recent works. The paper mainly compared the original KGW method. More recent and stronger baselines, including both token-level and semantic-level watermarking methods, should be compared for a more comprehensive evaluation.

3. Robustness evaluation is also limited. Stronger modification and paraphrasing attackers, beyond deletion, insertion, and synonym substitution, need to be presented to show the superiority of the proposed method against SOTA.

**Questions:**

See weaknesses.

---

> ### Author Response · Authors · 2025-11-29
> **Response to Reviewer xzV6 (1/3)**
>
> > Q1: The overhead of this method seems to be significant.
>
> We appreciate your practical concern. We would like to clarify that our method enables a controllable computation cost by parallel generation and kv-cache reuse. To assess the practical computation overhead, we report throughput and peak memory consumption across four datasets: Copen (short input / short output), ELI5 (short input / long output), HotpotQA (long input / short output), and MultiNews (long input / long output). All experiments are conducted on a single NVIDIA A800 GPU.
>
> | Generation Throughput | KGW  | Beam=2 | Beam=3 | Beam=4 | Beam=5 | Beam=6 |
> |-----------------------|------|--------|--------|--------|--------|--------|
> | Copen                 | 55.8 | 44.9   | 41.3   | 42.1   | 39.1   | 39.6   |
> | ELI5                  | 57.0 | 47.3   | 46.3   | 45.4   | 45.4   | 46.0   |
> | HotpotQA              | 23.3 | 11.6   | 8.3    | 5.3    | 4.1    | 3.9    |
> | MultiNews             | 54.7 | 46.5   | 34.9   | 31.1   | 29.8   | 23.6   |
>
> | Peak Memory Usage | KGW  | Beam=2 | Beam=3 | Beam=4 | Beam=5 | Beam=6 |
> |-------------------|------|--------|--------|--------|--------|--------|
> | Copen             | 12.6 | 12.8   | 12.9   | 13.0   | 13.1   | 13.2   |
> | ELI5              | 12.8 | 13.6   | 14.0   | 14.5   | 14.8   | 15.3   |
> | HotpotQA          | 14.9 | 21.2   | 24.7   | 28.1   | 31.5   | 35.0   |
> | MultiNews         | 14.9 | 21.8   | 25.8   | 30.2   | 32.6   | 36.7   |
>
> WaterSearch introduces additional cost, which is expected given its design. However, on short-input tasks such as Copen and ELI5, the impact on throughput is minimal and the increase in peak memory is modest. For long-input tasks, including HotpotQA and MultiNews, the overhead becomes more apparent; nonetheless, the growth remains sub-linear, demonstrating that WaterSearch scales efficiently in practice. Importantly, the method achieves substantial quality gains even at small parallel degrees (e.g., $k=2$, see Section 6.4), providing flexible hyper-parameters that can be tuned to accommodate different computational budgets. Please find the visualization, more results and discussion in Section 6.5 (highlighted in darkred).

---

> ### Author Response · Authors · 2025-11-29
> **Response to Reviewer xzV6 (2/3)**
>
> > Q2: The paper mainly compared the original KGW method. More recent and stronger baselines, including both token-level and semantic-level watermarking methods, should be compared for a more comprehensive evaluation.
>
> Thank you for this very practical question. We would like to clarify that this work proposes a general watermark searching and detection framework WaterSearch which could support different types of base watermarking methods. That means our framework is parallel and compatible to base watermarking methods, and could further improve the text quality of base method of searching in controllable computation cost. For detection, we could express the watermarked probability of each chunk in Eq. 10, and our document-level detection is expressed in Eq. 11, which could assess the watermarked confidence of each chunk and conclude an overall assessment.

---

> ### Author Response · Authors · 2025-11-29
> **Response to Reviewer xzV6 (3/3)**
>
> > Q3: Stronger modification and paraphrasing attackers, beyond deletion, insertion, and synonym substitution, need to be presented to show the superiority of the proposed method against SOTA.
>
> Thank you for your detailed and constructive comments. We further evaluate WaterSearch on 4 types of paraphrase attack. Here we report the experimental results for two classic paraphrasers, Pegasus and Parrot, as well as the corresponding bigram paraphrase attack following the settings of SimMark [1].
>
> | Method           | Pegasus | Pegasus-Bigram | Parrot | Parrot-Bigram | AVG  |
> |------------------|---------|----------------|--------|---------------|------|
> | KGW-Hard         | 79.4    | 67.6           | 7.5    | 9.2           | 40.9 |
> | WaterSearch  | 90.2 | 74.2           | 20.6   | 19.6          | **51.2** |
> | KGW-Soft         | 79.4    | 67.6           | 10.3   | 9.2           | 41.6 |
> | WaterSearch| 80.4    | 67.0           | 20.6| 18.6          | **46.7** |
>
> As shown in table above, our method outperforms its counterparts significantly, especially on the Parrot and Parrot-Bigram paraphrasers, achieving improvements of 10.3 and 9.4 percentage points over KGW-Soft. Please find full results and discussions in Section 6.3 and Appendix E.
>
> [1] Amirhossein et al., SimMark: A Robust Sentence-Level Similarity-Based Watermarking Algorithm for Large Language Models. EMNLP 2025.
>
> ---
> Again, we thank the reviewer for the valuable feedback. Please let us know if there are any other questions or suggestions.
>
> Best,
>
> Authors

---

### Official Review · Reviewer_7SBL · 2025-11-01

**Soundness:** 3
**Presentation:** 3
**Contribution:** 3
**Rating:** 6
**Confidence:** 3

**Summary:**

This paper proposes WaterSearch, a novel search-based framework for watermarking Large Language Model (LLM) outputs. The core idea is to move beyond token-level watermark embedding by generating multiple candidate text chunks in parallel and selecting the one that best balances text quality (fidelity to the original, unwatermarked model distribution) and watermark detectability (statistical strength of the watermark signal). The method is presented as a solution to the fundamental trade-off between these two objectives in existing watermarking schemes like KGW. The authors also introduce a new detection algorithm based on hypothesis testing and provide extensive experimental results showing significant improvements over strong baselines.

**Strengths:**

1. The shift from a purely token-level manipulation to a chunk-level search-and-select paradigm is a significant and compelling contribution. It elegantly reframes the watermarking problem as a multi-criteria optimization, which directly addresses a well-known limitation of existing methods.

2. The paper provides a theoretical analysis (Theorem 1) linking the macroscopic (sentence-level) selection objective with the microscopic (token-level) watermarking trade-off. This strengthens the methodological foundation and justifies the proposed approach.

3. The experiments are thorough and well-designed.
    *   Comprehensive Benchmarking: Evaluation across 10 diverse tasks and 3 major LLMs (Llama-2, InternLM, Qwen) demonstrates generalizability.
    *   Significant Performance Gains: The reported average improvement of 51.01% in downstream task performance under fixed detectability is impressive and clearly highlights the method's value.
    *   Robustness in Challenging Scenarios: The strong results on short-text (+47.78%) and low-entropy (e.g., code generation, +36.47%) scenarios are particularly noteworthy, as these are known pain points for current watermarks.
    *   Exceptional Attack Resilience: Maintaining high detectability under 80% word-level perturbations (deletion, insertion, substitution) is a remarkable result that significantly outperforms baselines.

**Weaknesses:**

1. Algorithm 1 requires generating $k$ candidate chunks in parallel at each step. This implies the generation time (latency) will be roughly $k$ times that of a baseline method. The paper's claim of "low computational cost" is misleading as it primarily focuses on memory (KV cache).
2. Algorithm 2 (Detection) appears to require the detector to "Recover the seeds from generation". This suggests the detector must know the exact context $c$ and the random seed generator used during generation. This is a much stronger assumption than KGW (which only needs a secret key) and may be fragile in black-box detection or if the context is slightly modified.

3.  The experiments fix $k$ (beam size) to 5 (1 vanilla + 4 watermarked). $k$ is a critical hyperparameter balancing quality, detectability, and latency, but the paper lacks a sensitivity analysis or ablation study on $k$.

**Questions:**

1.How exactly are the k−1 watermark seeds generated from context and recovered at detection time? Is the seed generator deterministic and keyed? What are attack consequences if this procedure is partially known?

2.Can you provide wall-clock runtime and peak GPU memory numbers for representative settings (e.g., k=5, chunk m=32) on a standard GPU? The asymptotic KV discussion is useful but practitioners will want absolute numbers.

3.Have you tried stronger/adaptive attackers (e.g., paraphrase-based sentence rewriting engineered to minimize green-token counts) or defenses that specifically target chunk-final tokens? If so, what happens to detectability?

4.For Theorem 1, can you relax the token-independence assumption or empirically show how well the mapping (f) holds across tasks?

---

> ### Author Response · Authors · 2025-11-29
> **Response to Reviewer 7SBL (1/3)**
>
> We thank the reviewer for the valuable feedback on improving this paper! Please find below our response to the reviewer’s questions.
>
> ---
> > Algorithm 2 (Detection) appears to require the detector to "Recover the seeds from generation". This suggests the detector must know the exact context c and the random seed generator used during generation. This is a much stronger assumption than KGW (which only needs a secret key) and may be fragile in black-box detection or if the context is slightly modified.
>
> > Q1 (part-1): How exactly are the k−1 watermark seeds generated from context and recovered at detection time? Is the seed generator deterministic and keyed?
>
> We appreciate your practical concern. The chunk-level seed generation process of WaterSearch is parallel to token-level text generation, and we would like to clarify that our method only requires a secret key as well. Unlike previous methods that use a secret key to manipulate distribution, we design a novel scheme to control a seed pool to enable diverse parallel generation. The process can be formalized as
>
> $Seed\\_Pool = Hash([s_1,s_2,\cdots],key)=shuffle([s_1,s_2,\cdots],key)$
>
> where $[s_1,s_2,\cdots]$ is a large group of seeds, and the hash function could be set to shuffle function condition of key.
> Besides, the token-level seed generation in the KGW framework can be expressed as:
>
> $Seed = Hash(x^{-h},\cdots,x^{-2},x^{-1})=(x^{-h}\cdot x^{-2}\cdot x^{-1})\mod M$
>
> where $seed$ is retrieved from a seed pool and $M$ is a large prime number [1][2]. This mechanism is widely discussed in [3] and here we set $h=1$ and align with KGW's settings.
> Below, we demonstrate the implementaion in simplified pseudocode:
> ```
> random.seed(41) # pre-defined scret key
> self.seed_pool = list(range(1, 5000)) # initialize seed pool
> for _ in range(L//m): # iteration over each chunk
>     random.shuffle(self.seed_pool) # build seed pool
>     cur_seeds = self.seed_pool[:self.K]
>     no_wm_chunk = model.generate(input_text_ids)
>     wm_chunks = []
>     for i in range(K):
>     # iteraction over each beam for search
>     # simplify for demonstration, achieve in parallel actually
>         token_seed = hash_func(input_text_ids[-h:], cur_seeds[i]) # build token-level seed
>         wm_chunk = model.generate(input_text_ids, wm_logit_processor, token_seed)
>         wm_chunks.append(wm_chunk)
>     idx = calcuate_best_wm_chunk(no_wm_chunk, wm_chunks) # calculate the best wm_chunk
>     input_text_ids = concat(input_text_ids, wm_chunks[idx])
>
> return tokenizer.decode(input_text_ids)
> ```
> It is worth notice that unlike token-level which modifies one token $x^{-1}$ could influence the detection accuracy in  future h steps $\\{x^1,\cdots,x^h\\}$, WaterSearch divides document into chunks and employs a $\chi^2$ statistic test to evaluate the probability of each chunk to return confidence globally, which makes token-level perturbation hard to influence detection accuracy.
>
> > Q1 (part-2): What are attack consequences if this procedure is partially known?
>
> Our seed generation process is controlled by a secret key to building the seed pool as discussed above (Eq. 12 in Appendix), which makes the attack hard to occur even when the procedure is fully known. Besides, unknown hyper-parameter settings like chunk length or beam num would make the attacker hard to simulate the generation process.
>
> We really appreciate the reviewer's suggestion. Please find pseudo-code of WaterSearch's detection process in Appendix A. Please find insertion, synonym substitution and paraphrase attack in Section 6.3 and Appendix E.
>
> [1] Kirchenbauer et al., A watermark for large language models. ICML 2023.
>
> [2] Fernandez et al., Three Bricks to Consolidate Watermarks for Large Language Models. IEEE WIFS, 2023.
>
> [3] Scott Aaronson and Hendrik Kirchner. Watermarking GPT outputs. 2023-02-01.

---

> ### Author Response · Authors · 2025-11-29
> **Response to Reviewer 7SBL (2/3)**
>
> > Q2: Can you provide wall-clock runtime and peak GPU memory numbers for representative settings (e.g., k=5, chunk m=32) on a standard GPU? The asymptotic KV discussion is useful but practitioners will want absolute numbers.
>
> Thank you for this very practical question. We report throughput and peak memory consumption across four datasets: Copen (short input / short output), ELI5 (short input / long output), HotpotQA (long input / short output), and MultiNews (long input / long output). All experiments are conducted on a single NVIDIA A800 GPU.
>
> | Generation Throughput | KGW  | Beam=2 | Beam=3 | Beam=4 | Beam=5 | Beam=6 |
> |-----------------------|------|--------|--------|--------|--------|--------|
> | Copen                 | 55.8 | 44.9   | 41.3   | 42.1   | 39.1   | 39.6   |
> | ELI5                  | 57.0 | 47.3   | 46.3   | 45.4   | 45.4   | 46.0   |
> | HotpotQA              | 23.3 | 11.6   | 8.3    | 5.3    | 4.1    | 3.9    |
> | MultiNews             | 54.7 | 46.5   | 34.9   | 31.1   | 29.8   | 23.6   |
>
> | Peak Memory Usage | KGW  | Beam=2 | Beam=3 | Beam=4 | Beam=5 | Beam=6 |
> |-------------------|------|--------|--------|--------|--------|--------|
> | Copen             | 12.6 | 12.8   | 12.9   | 13.0   | 13.1   | 13.2   |
> | ELI5              | 12.8 | 13.6   | 14.0   | 14.5   | 14.8   | 15.3   |
> | HotpotQA          | 14.9 | 21.2   | 24.7   | 28.1   | 31.5   | 35.0   |
> | MultiNews         | 14.9 | 21.8   | 25.8   | 30.2   | 32.6   | 36.7   |
>
> WaterSearch introduces additional cost, which is expected given its design. However, on short-input tasks such as Copen and ELI5, the impact on throughput is minimal and the increase in peak memory is modest. For long-input tasks, including HotpotQA and MultiNews, the overhead becomes more apparent; nonetheless, the growth remains sub-linear, demonstrating that WaterSearch scales efficiently in practice. Importantly, the method achieves substantial quality gains even at small parallel degrees (e.g., $k=2$, see Section 6.4), providing flexible hyper-parameters that can be tuned to accommodate different computational budgets.
> Please find the visualization, more results and discussion in Section 6.5 (highlighted in darkred).
>
> > Q3: Have you tried stronger/adaptive attackers (e.g., paraphrase-based sentence rewriting engineered to minimize green-token counts) or defenses that specifically target chunk-final tokens? If so, what happens to detectability?
>
> Thank you for your detailed and constructive comments. We further evaluate WaterSearch on 4 types of paraphrase attack. Here we report the experimental results for two classic paraphrasers, Pegasus and Parrot, as well as the corresponding bigram paraphrase attack following the settings of SimMark [4].
>
> | Method       | Pegasus | Pegasus-Bigram | Parrot | Parrot-Bigram | AVG  |
> |--------------|---------|----------------|--------|---------------|------|
> | KGW-Hard     | 79.4    | 67.6           | 7.5    | 9.2           | 40.9 |
> | WaterSearch  | 90.2    | 74.2           | 20.6   | 19.6          | **51.2** |
> | KGW-Soft     | 79.4    | 67.6           | 10.3   | 9.2           | 41.6 |
> | WaterSearch  | 80.4    | 67.0           | 20.6   | 18.6          | **46.7** |
>
> As shown in table above, our method outperforms its counterparts significantly, especially on the Parrot and Parrot-Bigram paraphrasers, achieving improvements of 10.3 and 9.4 percentage points over KGW-Soft. Please find full results and discussions in Section 6.3 and Appendix E.
>
> [4] Amirhossein et al., SimMark: A Robust Sentence-Level Similarity-Based Watermarking Algorithm for Large Language Models. EMNLP 2025.

---

> ### Author Response · Authors · 2025-11-29
> **Response to Reviewer 7SBL (3/3)**
>
> > Q4: For Theorem 1, can you relax the token-independence assumption or empirically show how well the mapping (f) holds across tasks?
>
> The token-independence assumption in Theorem 1 is used purely as an analytical simplification, specifically, to satisfy $\mathbb{E}\left[\frac{|\tilde{y}_G|}{|\tilde{y}|}\right]=\hat{P}_G$. WaterSearch itself or other watermarking methods do not rely on token independence—because our method operates at the chunk-level and preserves the model’s autoregressive structure.
>
> ---
> Again, we thank the reviewer for the valuable feedback. Please let us know if there are any other questions or suggestions.
>
> Best,
>
> Authors

---

### Note · Authors · 2025-12-02

I have read and agree with the venue's withdrawal policy on behalf of myself and my co-authors.